# An ESCRT-dependent step in fatty acid transfer from lipid droplets to mitochondria through VPS13D−TSG101 interactions

Jingru Wang[1,3], Na Fang[1,3], Juan Xiong[2,3], Yuanjiao Du[1], Yue Cao[1] & Wei-Ke Ji [1✉]

Upon starvation, cells rewire their metabolism, switching from glucose-based metabolism to mitochondrial oxidation of fatty acids, which require the transfer of FAs from lipid droplets (LDs) to mitochondria at mitochondria−LD membrane contact sites (MCSs). However, factors responsible for FA transfer at these MCSs remain uncharacterized. Here, we demonstrate that vacuolar protein sorting-associated protein 13D (VPS13D), loss-of-function mutations of which cause spastic ataxia, coordinates FA trafficking in conjunction with the endosomal sorting complex required for transport (ESCRT) protein tumor susceptibility 101 (TSG101). The VPS13 adaptor-binding domain of VPS13D and TSG101 directly remodels LD membranes in a cooperative manner. The lipid transfer domain of human VPS13D binds glycerophospholipids and FAs in vitro. Depletion of VPS13D, TSG101, or ESCRT-III proteins inhibits FA trafficking from LDs to mitochondria. Our findings suggest that VPS13D mediates the ESCRT-dependent remodeling of LD membranes to facilitate FA transfer at mitochondria-LD contacts.

[1] Department of Biochemistry and Molecular Biology, School of Basic Medicine, Tongji Medical College, Huazhong University of Science and Technology, Wuhan, Hubei, China. [2] Department of Anesthesiology, Tongji Hospital, Tongji Medical College, Huazhong University of Science and Technology, Wuhan, Hubei, China. [3]These authors contributed equally: Jingru Wang, Na Fang, Juan Xiong. ✉email: J_WK@hust.edu.cn

The fitness of an organism depends on its ability to adapt to changes in nutrient availability in the environment. To do so, cells inside organisms store energy as neutral lipid esters in lipid droplets (LDs), a type of lipid storage organelle conserved from yeast to humans, and then reprogram their metabolism from glucose catabolism to the mitochondrial oxidation of fatty acids (FAs) when nutrients become limited. Starvation triggers the release of free FAs from triacylglycerols stored in LDs and the transfer of FAs to the mitochondrial matrix for oxidation, which produces ATP to sustain cellular activities[1,2].

Although the biochemical basis of FA activation and import into mitochondria has been extensively studied for decades, our understanding of how FAs are transferred from LDs to mitochondria to drive FA oxidation under starvation is still limited. Current models propose that FA transfer relies on mitochondria-LD membrane contact sites (MCSs), where the membranes of the two opposing organelles are in close proximity (typically 10–30 nm)[3–6]. Consistent with this notion, the association of LDs and mitochondria increases in response to nutrient starvation in cultured cells[3,7–9] and exercise in skeletal muscle[10]. This spatial proximity between mitochondria and LDs can decrease the prevalence of cytosolic FAs, preventing lipotoxicity[3,8]. Another model suggests a role for mitochondria-LD MCSs in adipocytes, where mitochondria closely associated with large LDs exhibit reduced β-oxidation and display increased ATP synthesis, suggesting that the major function of these LD-associated mitochondria is to support the growth and expansion of LDs[11]. Therefore, it is plausible that mitochondria-LD MCSs serve as sites for both lipogenesis and lipolysis in different cell types or under different metabolic conditions[12–15]. Nevertheless, the molecular machinery that promotes FA transfer at mitochondria-LD MCSs remains poorly understood.

Vacuolar protein sorting-associated protein 13D (VPS13D) belongs to the metazoan VPS13 family. The human VPS13 family contains four members, namely, VPS13A, B, C, and D, whereas a single VPS13 exists in yeast[16]. A pioneering study demonstrated that VPS13A and VPS13C are lipid transporters at endoplasmic reticulum-mitochondria/LD and ER-late endosome/LD MCSs, respectively[17], while VPS13B is suggested to be a Golgi apparatus-associated protein required for Golgi integrity and neurite outgrowth[18,19]. Human VPS13 proteins are of great biomedical interest, as loss-of-function mutations in each of them are associated with genetic diseases[20–24]. Loss-of-function mutations of VPS13D cause a type of autosomal recessive ataxia with abnormal mitochondrial morphology, reduced energy generation, and lipidosis[23,24]. Human VPS13D is a giant protein consisting of 4363 amino acids (transcript variant 2, NM_018156), with a molecular weight of 493 kDa. VPS13D gene is the only essential gene in the VPS13 family, suggesting that VPS13D participates in cellular processes fundamental for organism survival[25]. A pioneering study identified VPS13D as a ubiquitin binding protein and a key regulator of mitochondrial size and clearance in *Dro*sophila[26].

Here we demonstrated that VPS13D coordinated FA trafficking in conjunction with the endosomal sorting complex required for transport (ESCRT) protein tumor susceptibility 101 (TSG101). The VPS13 adaptor-binding (VAB) domain of VPS13D and TSG101 directly remodels LD membranes. The lipid transfer domain of human VPS13D binds glycerophospholipids and FAs in vitro. Depletion of VPS13D, TSG101, or ESCRT-III proteins inhibits FA trafficking from LDs to mitochondria.

## Results

### The localization of mammalian VPS13D. To date, the localization and cellular functions of mammalian VPS13D remain

largely unknown. To explore the cellular functions of VPS13D, we initiated with an examination of the localization of endogenous VPS13D by immunofluorescence staining (IF). Although a commercial antibody was not initially effective in IF, its specificity for VPS13D could be improved by pre-clearing the antibody against fixed and permeabilized VPS13D-suppressed cells according to a protocol modified from a previous study[27] (Supplementary Fig. S1a). The efficiency and specificity of small-interfering RNAs (siRNAs)-mediated VPS13D suppression in the cells used for pre-clearing the antibody was confirmed by quantitative PCR (Fig. 1a) and western blotting (Fig. 1b). IF experiments using this pre-cleared antibody revealed that VPS13D was preferentially localized to the junctions between mitochondria and LDs in response to oleic acid (OA) stimulation in complete medium (CM) followed by 5 h of starvation in Earle's Balanced Salt Solution (OA/EBSS) (Fig. 1c, top panel); however, its localization at these junctions was almost completely lost upon VPS13D suppression (Fig. 1c, bottom panel). We found that 49% of endogenous VPS13D puncta were present at mitochondria-LD junctions (Fig. 1d), and 79% of these junctions were marked by VPS13D in wild-type cells (Fig. 1e).

To confirm the localization of VPS13D, we ectopically expressed a superfolder green fluorescent protein (sfGFP)-tagged VPS13D (transcript variant 2, NM_018156) (VPS13D^sfGFP), in which sfGFP was internally fused to VPS13D at a site shown to preserve yeast VPS13 function[28], in HEK293 cells. We used HEK293 cells for this purpose as they were well suited for transfection and expression of the huge VPS13D^sfGFP construct (~20 K base pairs in size). Consistent with the results of endogenous VPS13D IF experiments, a significant portion of VPS13D^sfGFP puncta was present at mitochondria-LD junctions in response to OA/EBSS by high-resolution confocal imaging (Supplementary Fig. 1b, c).

### Two amphipathic helices in the VPS13_C domain of VPS13D target LDs. The localization of VPS13D at mitochondria-LD junctions requires the presence of binding sites responsible for recognizing these two membranes. Based on bioinformatics analysis, VPS13D was found to consist of an N-terminal putative LTD, a UBA domain followed by a VAB domain, and a VPS13_C domain at its C terminus (Fig. 2a, top panel). To identify the regions responsible for recognizing LDs and mitochondrial membranes, respectively, we performed molecular dissection of VPS13D. Among the five truncated versions of VPS13D tested (tVPS13D1-5, Fig. 2a, bottom panel), a truncated version of the C-terminal region (tVPS13D-5) was associated with the LD membrane, as revealed by the co-localization between tVPS13D-5 and the class I LD membrane protein ACSL3 (Fig. 2b). Video analysis confirmed that the association of tVPS13D-5 with the LD membrane was stable and specific (Supplementary Fig. 2a). OA stimulation strongly enhanced the recruitment of tVPS13D-5 to the LDs (Fig. 2c). We further found that the VPS13_C domain alone (3981–4339 residues), which is present in tVPS13D-5, was exclusively localized to LDs in response to OA treatment (Fig. 2d), whereas under normal conditions, the VPS13_C domain could be recruited to both LDs and mitochondria (Supplementary Fig. 2b).

Previous studies have shown that the VPS13_C domain of VPS13A/C binds LDs through an amphipathic helix[17]. To examine whether the VPS13_C domain of VPS13D binds LDs by a similar mechanism, we searched for amphipathic helices in the VPS13_C domain using the HeliQuest tool[29]. Two amphipathic helices were identified, with the first helix (Helix-1) ranging from residues 3981 to 3998, while the other (Helix-2) ranging from residues 4050 to 4071 (Fig. 2e). To determine

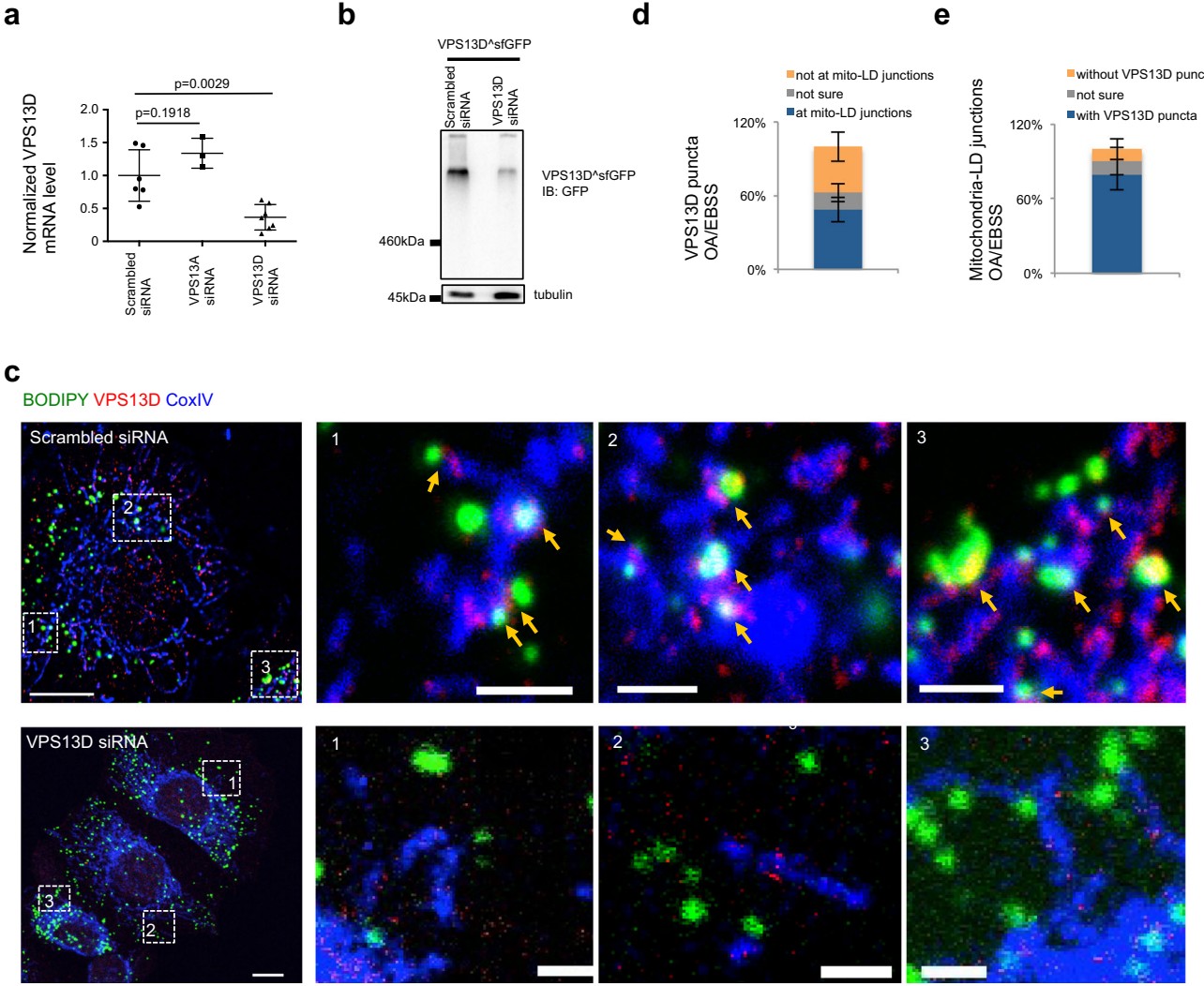

**Fig. 1 The localization of mammalian VPS13D. a** qPCR assays confirmed the efficiency and specificity of siRNAs-mediated VPS13D suppression in HEK293 cells. Two-tailed unpaired student's *t*-test. Mean ± SD. **b** Western blots demonstrated the efficiency of siRNAs-mediated VPS13D suppression in HEK293 cells transfected with VPS13D^sfGFP with anti-GFP and anti-tubulin antibody. **c** Confocal image of fixed HEK293 cells stained with pre-cleared anti-VPS13D antibody (red), BODIPY493/503 (LD; green), and anti-CoxIV antibody (mitochondria; blue) upon treatments with either scrambled (top panel) or VPS13D siRNAs (bottom panel). Left: merged image of whole cells; right: three insets from boxed regions in whole cell image. Contrast range of images was set to the same level for both scrambled and VPS13D siRNAs treated cells. Yellow arrows denoted VPS13D puncta at potential mitochondria-LD junctions. Four independent imaging sessions were performed with similar results. **d** Percentage of VPS13D puncta (n = 644 from 20 cells in four independent assays) present at mitochondria-LD junctions upon OA/EBSS treatment. Mean ± SD. **e** Percentage of mitochondria-LD junctions (n = 286 from 20 cells in four independent assays) marked with VPS13D puncta upon OA/EBSS treatment. Mean ± SD. Scale bar, 10 μm in whole cell image and 2 μm in insets in **c**.

whether these two amphipathic helices were required for VPS13_C domain binding to LDs, point mutations were introduced into the hydrophobic interfaces of each helix (Helix-1: L3991Q; Helix-2: L4052Q and L4053Q) to break their structures (Fig. 2e). Disruption of both Helix-1 and Helix-2 not only blocked the binding of VPS13_C to LDs under normal conditions (Supplementary Fig. 2c) but also in response to OA stimulation (Fig. 2f), indicating that the binding of VPS13D to LDs is mediated through the two amphipathic helices in the VPS13_C region.

We further tested whether the two helices were required for LD binding of full-length VPS13D. VPS13D with three point mutations in the two helices (L3991Q L4052Q L4053Q) failed to associate with LDs. Instead, this mutant associated with mitochondria which appeared to be fragmented (Fig. 2g).

**The N-terminal region of VPS13D targets mitochondria**. To determine which region of VPS13D is required to target mitochondria, we generated a set of N-terminal VPS13D truncations (Fig. 3a). A region at the N-terminal portion of VPS13D (residues 404–913) showed mitochondrial localization in response to OA treatment (Fig. 3b, bottom panel), whereas mitochondrial localization was barely detectable by microscopy without OA treatment (Fig. 3b, top panel). We found that a region of VPS13D containing residues 404-612 was sufficient to target the outer mitochondrial membrane (OMM) in an OA-dependent manner (Fig. 3c), whereas the other region from residues 613-913 failed to localize to the OMM (Supplementary Fig. 2d, e). This region of VPS13D (residues 404–612) is referred to as VPS13D_N hereafter. Cell fractionation experiments confirmed the enrichment of VPS13D_N in crude mitochondrial fractions upon OA

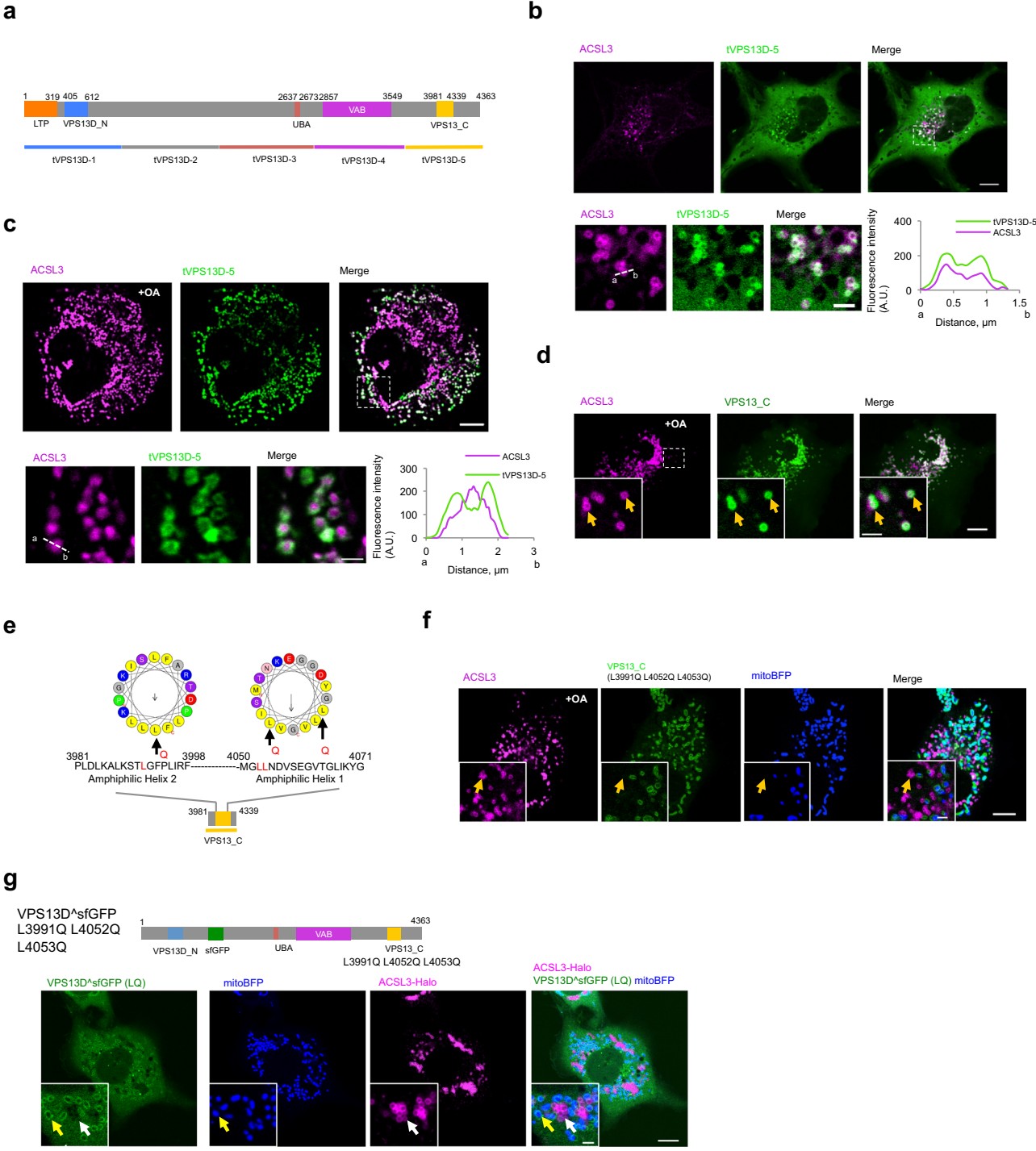

**Fig. 2 Two amphipathic helices in the VPS13_C domain target LDs. a** Diagram of the domain organization of VPS13D (top panel) and five truncated versions (bottom panel). **b** Confocal image of a COS7 cell expressing GFP-tVPS13D-5 (green) and ACSL3-Halo (magenta) in CM. Top: whole cell image; Bottom: one inset from boxed region in whole cell image with linescan analysis (a->b) on the right. A.U., arbitrary unit. **c** Confocal image of a COS7 cell as in **b** under OA treatment. A.U., arbitrary unit. **d** Confocal image of a COS7 cell expressing GFP-VPS13_C (green) and ACSL3-Halo (magenta) under OA treatment. Yellow arrows denoted VPS13_C decorated LDs. **e** Diagram of two predicted amphipathic helical arrangements of residues 3,981-3,998 (helix-1) and residues 4,050-4,071(helix-2) of the VPS13_C domain generated via the HeliQuest tool[29] (v1.2; heliquest.ipmc.cnrs.fr). Three point mutations (L -> Q, black arrows) were introduced to the hydrophobic interfaces of these two helices. **f** Confocal image of a COS7 cell expressing GFP-VPS13_C mutant (green, L3991Q in helix-1; L4052Q, L4053Q in helix-2) under OA treatment. Yellow arrows indicated LDs without VPS13_C. **g** Confocal image of a HEK293 cell expressing VPS13D^sfGFP (L3991Q L4052Q L4053Q) (green), mitoBFP (blue) and ACSL3-Halo (magenta). White arrows indicated LDs without VPS13D^sfGFP (L3991Q L4052Q L4053Q) and yellow arrows denoted mitochondrially associated VPS13D^sfGFP (L3991Q L4052Q L4053Q). Scale bar, 10 μm in whole cell image and 2 μm in insets in **b**, **c**, **d**, **f** and **g**.

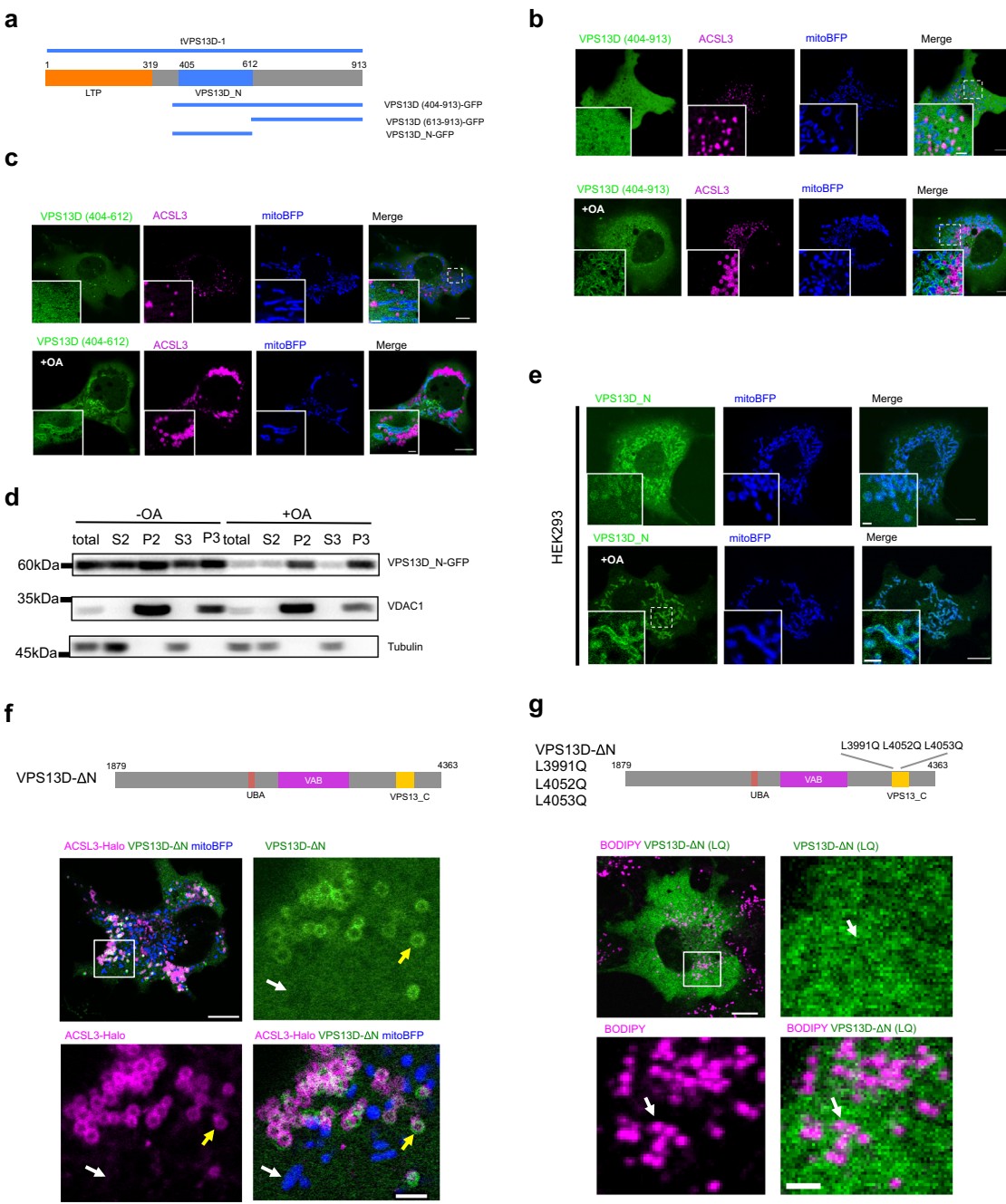

**Fig. 3 The VPS13D N-terminal region targets mitochondria. a** Diagram of truncation mutants of the VPS13D N-terminal region. **b** Confocal image of a COS7 cell expressing VPS13D (404-913)-GFP in CM without OA (top panel) or with OA (bottom panel). **c** Confocal image of a COS7 cell stably expressing VPS13D_N-GFP (green), mitoBFP (blue) and ACSL3-Halo (magenta) in CM without OA (top panel) or with OA (bottom panel). One inset from a boxed region was shown at bottom left. **d** Western blots demonstrated the distribution of the VPS13D_N-GFP protein in subcellular fractions in COS7 cells as in **c** S2/ P2: 7000 *g* supernatant and pellet; S3/P3: 20,000 *g* supernatant and pellet. **e** Confocal image of a HEK293 cell expressing VPS13D_N-GFP in CM without OA (top panel) or with OA (bottom panel). One inset from a boxed region was shown at bottom left. **f** Confocal image of a COS7 cell expressing GFP-VPS13D-ΔN in CM without OA. Yellow arrows indicated LDs with GFP-VPS13D-ΔN while white arrows denoting mitochondria not associating with GFP-VPS13D-ΔN. **g** Confocal image of a COS7 cell expressing GFP-VPS13D-ΔN (L3991Q L4052Q L4053Q) in CM without OA. White arrows indicated the LDs not associating with GFP-VPS13D-ΔN (L3991Q L4052Q L4053Q). Scale bar, 10 μm in whole cell image and 2 μm in insets in **b**, **c**, **e**, **f** and **g**.

stimulation in COS7 cells stably expressing VPS13D_N (Fig. 3d). Mitochondrial localization of VPS13D_N was further confirmed in HEK293 cells by confocal microscopy. A fraction of VPS13D_N was localized to mitochondria even in the absence of OA (Fig. 3e, top panel), and its recruitment to the OMM was enhanced upon OA stimulation (Fig. 3e, bottom panel).

The VPS13D mutant with a deletion of the N-terminal region (VPS13D-Δ1–1878) showed no association with mitochondria (Fig. 3f), indicating that the N-terminal region is required for mitochondrial targeting. Instead, VPS13D-Δ1–1878 associated with LDs (Fig. 3f), and disruption of the two amphipathic helices in VPS13_C by introducing the three point mutations (L3991Q

L4052Q L4053Q) to VPS13D-Δ1–1878 blocked its LD localization (Fig. 3g).

**The VAB domain of VPS13D interacts with TSG101.** Next, we investigated the cellular localization and functions of the VAB domain of VPS13D. The VAB domain with a GFP tagged at its C-terminus (VAB-GFP) formed bright puncta associating with LDs and mitochondria upon OA/EBSS stimulation in HEK293 cells (Supplementary Fig. 3a, b). We noted that the VAB domain was preferentially localized to one side of each LD instead of being distributed over the entire LD surface (Supplementary Fig. 3a, c). Time-lapse images confirmed that the association of VAB puncta with LDs (Supplementary Fig. 3d) and mitochondria (Supplementary Fig. 3e) was stable. The VAB domain contains six repeats (R1–R6), which are conserved in four human VPS13 proteins[30]. Expression of either R1–R3 or R4–R6 alone resulted in diffused localization all over the cell without the formation of puncta (Supplementary Fig. 3f, g), suggesting that all six repeats (R1–R6) were required for VAB-GFP localization. Cell fractionation analysis demonstrated that the majority of VAB-GFP was found in the cytosolic fractions, whereas its presence in the crude mitochondrial fraction (P2/P3) could be detected upon Dithiobis (succinimidyl propionate)-mediated protein−protein cross-linking (Supplementary Fig. 3h), suggesting that the association of VAB-GFP with LDs or mitochondria is transient and might be mediated by protein−protein interactions.

We found a Pro-Ser-Ala-Pro (PSAP, residues 3044–3047) motif in the VAB domain of VPS13D (Fig. 4a), which was highly conserved in mammals and specific to VPS13D rather than other mammalian VPS13 paralogs (Supplementary Fig. 4a). The PS/TAP motif was previously identified in HIV-1, the Ebola virus, and other enveloped viruses that can recruit TSG101, an ESCRT protein[31,32]. The ESCRT complex plays essential roles in multiple core cellular events involving membrane remodeling[33,34], such as cytokinesis[32], endosome maturation[35,36], autophagy[37], membrane repair[38–41], and FA trafficking from LDs to peroxisomes[42]. The presence of a PSAP motif in the VAB domain of VPS13D suggested a potential interaction between VPS13D and TSG101. Indeed, GFP-Trap assays confirmed the interaction between VPS13D and TSG101 in HEK293 cells (Supplementary Fig. 4b), in agreement with a previous study on *Drosophila*[26]. The VAB domain was sufficient for VPS13D interactions with both endogenous TSG101 and Halo-TSG101 (Fig. 4b). The interaction between VAB domain and endogenous TSG101 was also confirmed in COS7 cells stably expressing VAB-GFP (Supplementary Fig. 4c). Deletion of the PSAP sequence in the VAB domain reduced its interaction with TSG101 by ~60% (Fig. 4c), as assessed by GFP-Trap assays, but did not entirely eliminate the interaction, indicating that this motif has an important role in the VAB–TSG101 interaction but also suggesting that additional regions of the VAB domain and TSG101 interact. TSG101 protein contains several unique domains, such as an inactive ubiquitin-conjugating domain (UEV), a proline-rich (PR) region, a coiled-coil (CC) domain, and a steadiness box (SB) (Fig. 4d)[43]. The UEV domain and PR-CC-SB domains independently interacted with the VAB domain, as revealed by GFP-Trap assays (Fig. 4e, Supplementary Fig. 4d), indicating a bimodal interaction between the VAB domain and TSG101 in a manner reminiscent of TSG101 interaction with the ESCRT-0 protein, namely, hepatocyte growth factor-regulated tyrosine kinase substrate (HGS)[44] or a TSG101-associated ligase Tal[45]. For instance, the UEV domain of TSG101 was found to independently interact with the PR domain and C-terminal clathrin-binding domain of HGS[44]. Although TSG101 was previously found to interact with HGS on endosomal membranes[44], we found that VAB−GFP puncta did

not associate with HGS-decorated endosomal membranes under OA/EBSS treatment (Supplementary Fig. 4e, f), suggesting that the VAB−TSG101 interaction is independent of HGS.

Next, we investigated the cellular localization of TSG101. Previous studies have shown that TSG101 preferentially localizes to the late endosomal structures, and that its cellular localization is under cell cycle regulation[46,47]. To avoid the artifacts due to high levels of Halo-TSG101 expression driven by the cytomegalovirus promoter (CMV), we generated a stable HEK293 cell line that was infected with a lentiviral vector expressing the transgene under the control of a weak promoter EF1α; this cell line showed a much lower and more homogeneous level of Halo-TSG101 among cells (hereafter referred to as Halo-TSG101), as revealed by flow cytometry analysis (Supplementary Fig. 4g). TSG101 was not localized to LDs under normal conditions (Supplementary Fig. 4h, i), whereas a significant portion (~25%) of Halo-TSG101 puncta was recruited to LDs after OA/EBSS stimulation (Fig. 4f, g). In addition, the recruitment of endogenous TSG101 to LDs was confirmed in OA/EBSS-treated HEK293 cells by IF staining with TSG101 antibody, the specificity of which was validated by siRNAs-mediated suppression in IF experiments (Supplementary Fig. 5a–d).

Next, we sought to determine the role of VPS13D in TSG101 localization to LDs. Depletion of VPS13D abolished the recruitment of TSG101 to LDs under OA/EBSS stimulation (Fig. 4f, g). In contrast, overexpression of VAB-GFP increased TSG101 recruitment to LDs upon OA/EBSS treatment by ~3.2-fold. Deletion of PSAP significantly reduced the TSG101 recruitment resulting from VAB-GFP overexpression (Fig. 4f, g and Supplementary Fig. 5e, f). The UEV domain alone and PR-CC-SB domain alone failed to associate with LDs in response to VAB-GFP overexpression under OA/EBSS stimulation (Supplementary Fig. 5g, h), suggesting that the bivalent interaction of VAB and TSG101 might be critical for TSG101 recruitment to LDs.

One of the most noticeable features of VAB/TSG101-decorated LDs was that the morphology of LDs was dramatically modified; we observed both constriction of LDs (Fig. 4h, insets 1, 3, 4, 6) and budding/tubulation from LDs (Fig. 4h, insets 2, 5, 6) with specific co-localization of VAB-GFP and Halo-TSG101 at constricted/budding sites, as shown by high-resolution confocal live cell microscopy. This suggests that TSG101 and the VAB domain of VPS13D have direct roles in modulating LD morphology. To further confirm the roles of the VAB domain of VPS13D and TSG101 in the remodeling of LD membranes, we directly examined the LD morphologies upon OA/EBSS stimulations by performing transmission electron microscopy (TEM) of HEK293 cells co-expressing VAB-GFP and Halo-TSG101. Electron micrographs revealed that a substantial portion of LDs was dramatically deformed upon overexpression of Halo-TSG101 and VAB-GFP (Fig. 4i, j): LDs underwent either budding (Fig. 4i, right panel; inset 1) or constriction (Fig. 4i, right panel; inset 2). However, the LDs in OA/EBSS-treated cells expressing the empty vectors were typically spherical with mild deformations (Fig. 4i, left panel). In addition, we noted that the mitochondria were highly condensed in the VAB/TSG101-overexpressing cells, suggesting that the mitochondrial functions might be impaired in these cells.

Given the dynamic nature of mitochondria and LDs, we tracked the VAB/TSG101-decorated LDs relative to mitochondria over time and found that the VAB/TSG101 could be specifically localized at mitochondria−LD contacts, and that the size of BODIPY-C12-labeled LDs appeared to gradually decrease over time as they contacted mitochondria (Fig. 4k and Supplementary Movie 1). Collectively, our findings suggest that VPS13D and TSG101 directly remodel LD membranes in a cooperative manner.

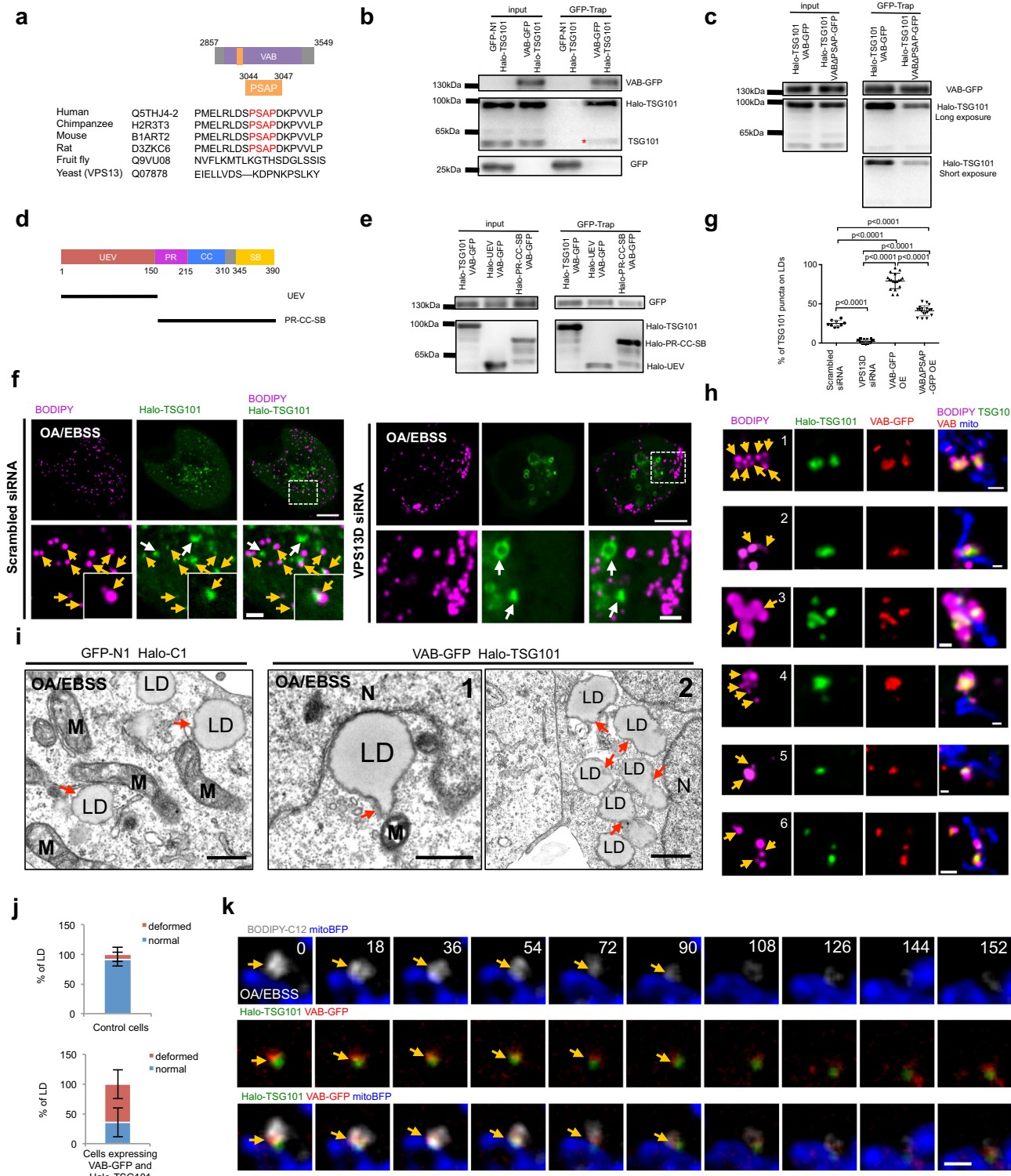

**VPS13D is required for maintenance of mitochondria–LDs interactions**. The presence of VPS13D at mitochondria−LD junctions upon OA/EBSS treatment led us to hypothesize that VPS13D might play a role at such MCSs. Therefore, we initially explored the role of VPS13D in the maintenance of mitochondria −LD MCSs using Doxycycline (Dox)-inducible mitochondria −LD MCSs reporter HEK293 cell lines, in which the proximity-mediated GFP reconstitution of Tom20-GFP$_{1-10}$ and Plin2-GFP$_{11}$ specifically occurs at mitochondria−LD MCSs. Because the reconstitution of GFP was irreversible, a low concentration (0.1

µM) of Dox and short incubation time (12 or 18 h) were applied to avoid artificial tethering between mitochondria and LDs. Under this condition, the morphology of both mitochondria and LDs appeared to be normal, and GFP signals were specifically confined to mitochondria−LD MCSs (Fig. 5a). Suppression of VPS13D substantially decreased the formation of mitochondria −LD MCSs in response to OA/EBSS treatment by confocal microscopy (Fig. 5b) and by high-throughput flow cytometry (Fig. 5c and Supplementary Fig. 6a-c). Immunoblot analysis showed that the expression level of GFP$_{1-10}$ was not significantly

**Fig. 4 The VAB domain of VPS13D interacts with TSG101. a** Alignment of PSAP motifs of VAB domains of VPS13 proteins. **b** GFP-Trap assays showing interactions between VAB-GFP and Halo-TSG101 with anti-GFP and anti-TSG101 antibody. The band of endogenous TSG101 was indicated by a red asterisk. **c** GFP-Trap assays comparing the interactions between VAB-GFP/Halo-TSG101 and VAB-ΔPSAP-GFP/TSG101 with anti-GFP antibody and anti-TSG101 antibody. **d** Schematic diagram of TSG101 domain organization. **e** GFP-Trap assays showing both Halo-UEV and Halo-PR-CC-SB interacting with Halo-TSG101 with anti-GFP and anti-Halo antibody. **f** Confocal image of HEK293 cells labeled with Halo-TSG101 (green) and BODIPY 558/568 (magenta) upon treatments with either scrambled (left panel) or VPS13D siRNAs (right panel) under OA/EBSS stimulation. Yellow arrows denoted TSG101 puncta associating with LDs while white arrows indicated TSG101 puncta not on LDs. **g** Percentage of TSG101 decorated LDs in scrambled ($n = 438$ from 9 cells), VPS13D siRNAs ($n = 281$ from 17 cells) treated, VAB overexpressed ($n = 298$ from 18 cells) or VAB-ΔPSAP overexpressed cells ($n = 312$ from 18 cells). Three independent imaging sessions were performed with similar results. Ordinary one-way ANOVA (Tukey's multiple comparisons test). ****$p < 0.0001$. Mean ± SD. **h** High-resolution confocal images of HEK293 cells expressing Halo-TSG101 (green), VAB-GFP (red), mitoBFP (blue) and labeled with BODIPY 558/568 (magenta) under EBSS/OA stimulation. Yellow arrows denoted LD remodeling with VAB and TSG101 specific enrichment at constrictive/tubulating sites. **i** TEM micrographs of LDs in OA/EBSS-treated HEK293 cells either expressing empty vector (left panel) or co-expressing VAB-GFP and Halo-TSG101 (right panel). M: mitochondria; N: nucleus. Red arrows denoted sites of LD budding/tabulation (right panel; inset 1) or constrictions (right panel; inset 2). **j** Percentage of deformed LDs in control ($n = 89$ from 19 cells) or cells co-expressing VAB-GFP and Halo-TSG101 ($n = 76$ from 17 cells). Three biological replicates were performed with similar results. Mean ± SD. **k** Time-lapse images of an OA/EBSS-treated HEK293 cell as in **h** with yellow arrows denoting VAB /TSG101 specific enrichment at mitochondria-LDs MCSs. Time in sec. Scale bar, 10 μm in whole cell image and 2 μm in insets in **g**; 1 μm in **h**; 0.5 μm in **i**; 2 μm in **k**.

altered upon VPS13D suppression (Supplementary Fig. 6d and e). In addition, the effects of VPS13D suppression on mitochondria−LD MCSs were independent of the extent of Dox induction, as shown by flow cytometry analysis under milder Dox induction (0.1 μM, 12 h) (Supplementary Fig. 6f–h).

We further confirmed the role of VPS13D in the maintenance of mitochondria-LD junctions by using a reversible dimerization-dependent fluorescent proteins (ddFPs) system[48]. We observed that the ddGFP signals were specifically enriched at mitochondria–LDs junctions but these signals were much weaker compared with the splitGFP signals, possibly because ddGFP dimerization is reversible (Supplementary Fig. 6i). Suppression of VPS13D significantly reduced the mitochondria–LD interaction but TSG101 depletion did not substantially alter such MCSs, as revealed by flow cytometry analysis of ddGFP fluorescence (Supplementary Fig. 6j).

In addition, we examined the effects of VPS13D suppression on mitochondria–LD interactions by co-localization analysis. Suppression of VPS13D substantially reduced the extent of mitochondria−LD interactions in HEK293 cells (Fig. 5d, e), as revealed by the decreased number of co-localized pixels between mitoBFP and ACSL3-Halo. In addition, VPS13D suppression resulted in more LDs at the cell periphery and fewer LDs at perinuclear regions (Fig. 5f). VPS13D depletion led to a significant increase in both LD size (Fig. 5g) and density (number of LDs / cell area) (Fig. 5h). Moreover, the duration of mitochondria−LD contacts decreased upon VPS13D suppression, as determined by live-cell imaging (Fig. 5i, j and Supplementary Movie 2).

**VPS13D and TSG101 are required for efficient transfer of FAs from LDs to mitochondria.** Given that the transfer of FAs from LDs to mitochondria may rely on mitochondria-LD MCSs, we next explored whether VPS13D plays a role in the transfer of FA using fluorescent FA analog BODIPY-C12 (Red C12)-based pulse-chase assays as previously described[3] (Supplementary Fig. 7a). Red C12 was initially incorporated into LDs ($t = 0$) and was gradually transferred from LDs to mitochondria after starvation ($t = 5$ or 24 h), as revealed by the presence of Red C12 fluorescence in mitochondria (Fig. 6b). In addition, we found that the transfer rate of Red C12 from LDs to mitochondria contacting LDs was significantly higher than that to mitochondria not adjacent to LDs by live cell microscopy after 5 h of starvation (Supplementary Fig. 7b, c).

We next use RNA interference to determine the role of VPS13D in FA transfer from LDs to mitochondria. VPS13D suppression substantially inhibited the transfer of Red C12 to

mitochondria upon starvation, as revealed by the large amount of Red C12 that was still incorporated in LDs, while less was trafficked to mitochondria (Fig. 6a, b, k). Both the size and number of Red C12-labeled LDs markedly increased after 5 h of starvation in VPS13D-suppressed cells compared with control cells (Fig. 6a, b, i and j).

We further investigated whether TSG101 and other components of the ESCRT complex were required for FA trafficking. Distinct from a canonical role of ESCRTs that function in inverse budding reactions on negatively curved membranes, four ESCRT proteins, namely, CHMP6, CHMP4B, CHMP1B, and IST1, were previously suggested to mediate classical-topology membrane shaping reactions on positively curved membranes[34,42,49], which more likely fits into the scenario of remodeling of LDs membranes during FAs trafficking. Therefore, we performed siRNA-based analysis of the four ESCRT proteins as well as TSG101 and ALIX, an ESCRT-associated protein required for cytokinesis[32], using Red C12 pulse-chase assays in cells starved for 5 h. Suppression of TSG101 or CHMP4B resulted in a ~3-fold increase in the size of red C12-labeled LDs after 5 h of starvation (Fig. 6c, f, i). Moreover, depletion of VPS13D, TSG101, CHMP4B, CHMP6, or IST1 led to a significant increase in the number of Red C12-labeled LDs compared with control cells (Fig. 6a–i, j). In addition to VPS13D, depletion of TSG101, CHMP6, or the ESCRT-III proteins resulted in a significant reduction of Red C12 transfer to mitochondria (Fig. 6c-g, k), suggesting that these ESCRT proteins play important roles in FA trafficking from LDs to mitochondria. In contrast, depletion of ALIX did not significantly affect either the number or the size of red C12-labeled LDs, or the transfer of FA upon 5 h of starvation (Fig. 6h, i, j & k), suggesting that ALIX might be dispensable for this process. In addition, we found no substantial changes in the total amounts of BODIPY-C12 taken up by either control, VPS13D, or ESCRT protein-depleted cells at $t = 0$ (right before starvation) (Fig. 6l), suggesting that the increase in the size and number of LDs was not due to more FAs taken up by VPS13D, or ESCRT proteins-suppressed cells. Taken together, our findings suggested that VPS13D and the ESCRT complex were required for efficient FA trafficking from LDs to mitochondria. However, it should be noted that the fluorophore-conjugated FAs likely have different biophysical properties from the endogenous FAs, which could affect on their trafficking and metabolism[50], the in vivo FA trafficking results using BODIPY-conjugated FAs should be further verified with the use of isotopically labeled FAs.

As an alternative approach, we confirmed the roles of VPS13D and TGS101 in the transport and oxidation of endogenous FAs in

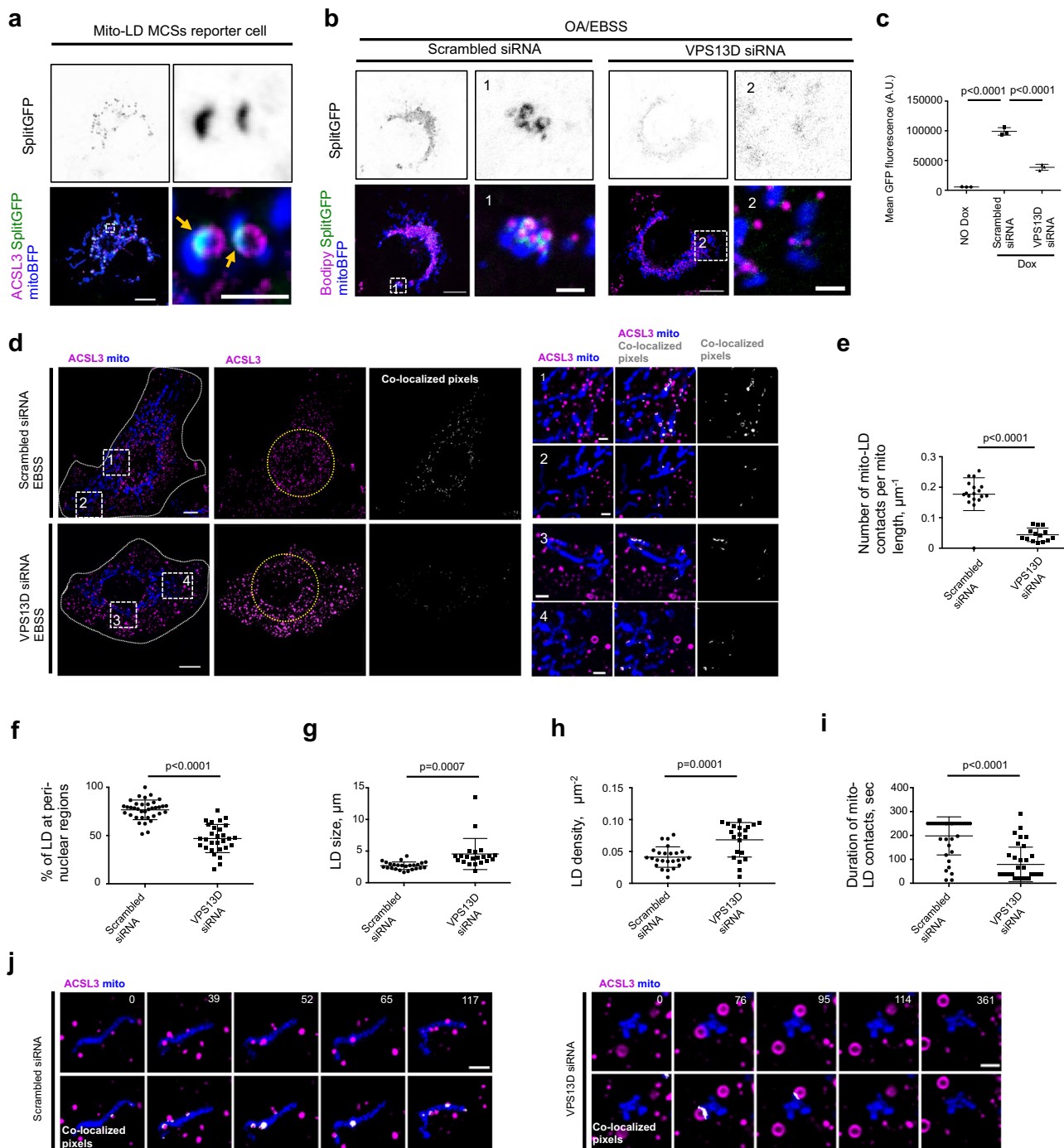

**Fig. 5 VPS13D was required for maintenance of mitochondria-LDs interactions. a** Confocal image of a Dox-inducible mitochondria-LDs MCSs reporter cell expressing mitoBFP (blue) and ACSL3-Halo (magenta). Yellow arrows denoted the reconstituted GFP (green) at MCSs. SplitGFP representing the reconstituted GFP fluorescence at mitochondria-LD junctions. **b** Cells as in **a** were treated with either scrambled or VPS13D siRNAs under OA/EBSS. Left: whole cell image; Right: one inset from a boxed region in whole cell image. **c** Quantification of mean GFP fluorescence intensity of mitochondria-LDs MCSs reporter cells upon treatment with either scrambled or VPS13D siRNAs by flow cytometry. A.U., arbitrary unit. **d** Confocal image of HEK293 cells expressing ACSL3-Halo (magenta) and mitoBFP (blue) upon treatments with scrambled (top panel) or VPS13D siRNAs (bottom panel). Left: whole cell image; Right: two insets from boxed regions in whole cell image with co-localized pixels in white representing MCSs. **e** Quantification of number of mitochondria-LDs MCSs per mitochondrial length; 18 scrambled siRNA treated cells and 14 VPS13D siRNA treated cells were calculated. **f** Percentage of LDs at peri-nuclear region; 36 scrambled siRNA treated cells and 29 VPS13D siRNA treated cells were calculated. The peri-nuclear region was defined as 10 μm distal to rims of nucleus. **g, h** Quantification of LDs size (**g**) and density (**h**); 25 scrambled siRNA treated cells and 21 VPS13D siRNA treated cells were calculated. **i** Duration of mitochondria-LDs interactions. 35 ROIs with well-resolved mitochondria and LDs from scrambled siRNA treated cells and 35 ROIs from VPS13D siRNA treated cells were calculated. **j** Representative time-lapse images of cells as in **d** with white pixels representing possible mitochondria-LDs junctions. At least three independent imaging sessions were performed with similar results in **c, e**–**i**. The statistic analysis was performed with two-tailed unpaired student's *t*-test, and data are means ± SD in **c, e**–**i**. Time in sec. Scale bar, 10 μm in whole cell image and 2 μm in insets in **a, b, d** and 2 μm in **j**.

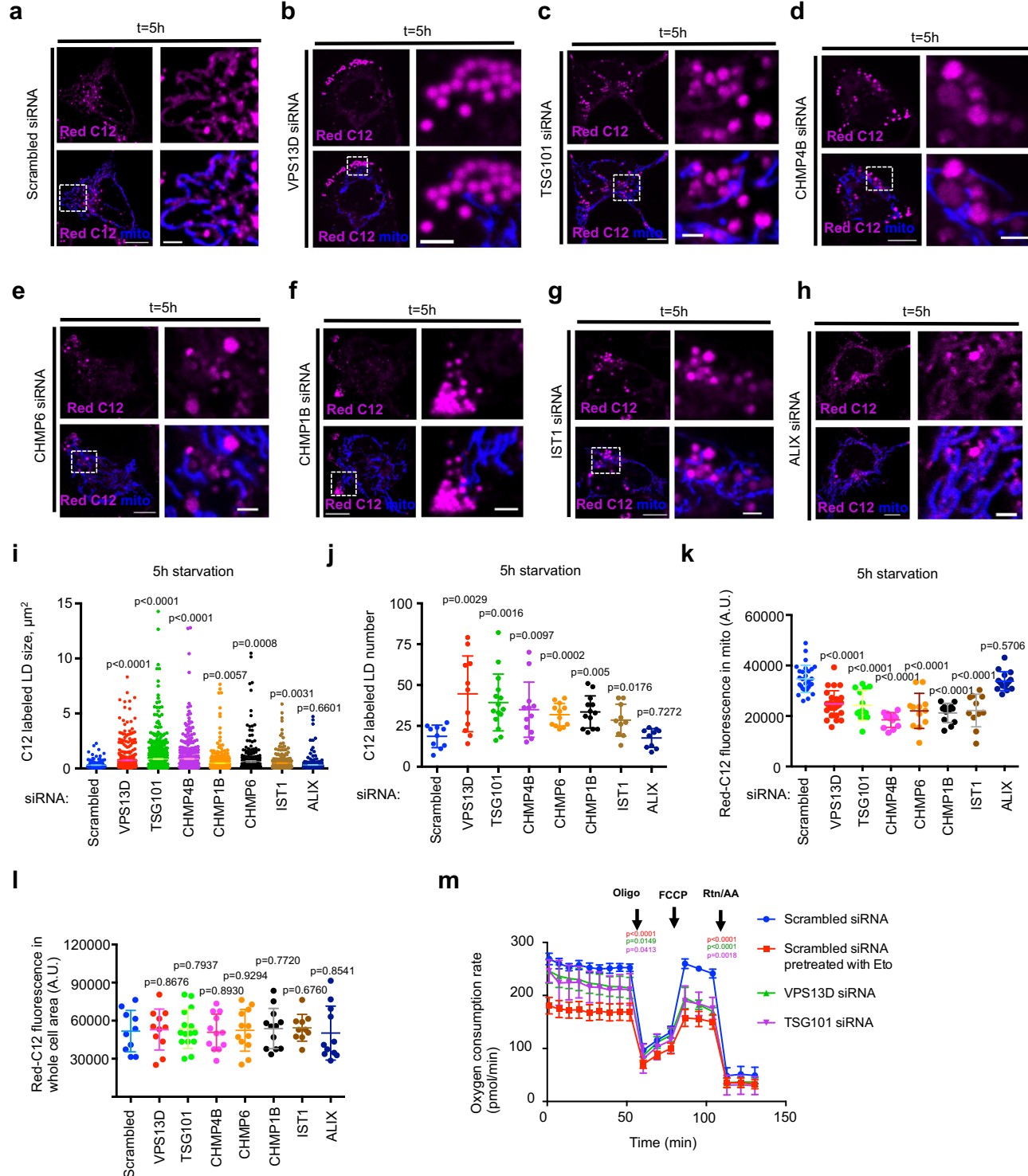

mitochondria via seahorse FA oxidation (FAO) assays. siRNA-mediated suppression of either VPS13D or TSG101 in HepG2 cells cultured in substrate-limited medium led to a slight reduction in basal respiration but a higher reduction in the maximal respiration of mitochondria. As a control, cells were pretreated with Etomoxir (Eto, a carnitine palmitoyl transferase-1 inhibitor), which substantially reduced both the basal and maximal respiration of mitochondria (Fig. 6m). The reduced FA oxidation observed in VPS13D-suppressed cells was consistent with the findings of two clinical studies, in which cells from patients of ataxia with spasticity (caused by loss-of-function mutations of VPS13D) showed lipidosis and reduced energy productions[23,51]. In addition, a previous study showed that the cristae of mitochondria appeared to be intact in VPS13D-depleted cells, as observed by TEM, indicating that the capacity of mitochondria for energy generation might be largely unaffected upon VPS13D-suppression[26]. Therefore, these results suggest that the low energy production and lipodosis observed in the patient cells may be attributed to the impaired transfer of FAs to mitochondria.

**Fig. 6 VPS13D and TSG101 are required for efficient FAs transfer from LDs to mitochondria. a–h** Confocal image of HEK293 cells in Red C12 pulse-chase assays after treatments with scrambled (**a**), VPS13D (**b**), TSG101 (**c**), CHMP4B (**d**), CHMP6 (**e**), CHMP1B (**f**), IST1 (**g**), or ALIX siRNAs (**h**). **i**, **j** Quantification of size (**i**) and number (**j**) of Red C12-labeled LDs after 5 h of starvation. Scrambled ($n = 177$ from 10 cells), VPS13D ($n = 488$ from 11 cells), TSG101 ($n = 589$ from 15 cells), CHMP4B ($n = 418$ from 12 cells), CHMP1B ($n = 475$ from 12 cells), CHMP6 ($n = 322$ from 13 cells), IST1 ($n = 284$ from 10 cells) or ALIX ($n = 257$ from 10 cells) siRNAs treated cells were analyzed. **k** Relative rate of Red C12 uptake to mitochondria upon 5 h of starvation was quantified by measurement of C12 fluorescence inside mitochondria. Scrambled (30 cells), TSG101 (14 cells), CHMP4B (12 cells), CHMP1B (12 cells), CHMP6 (12 cells), IST1 (11 cells) or ALIX (15 cells) siRNAs treated cells were analyzed. A.U., arbitrary unit. **l** Quantification of Red C12 fluorescence in whole cell areas. Scrambled (10 cells), VPS13D (11 cells), TSG101 (15 cells), CHMP4B (12 cells), CHMP1B (12 cells), CHMP6 (12 cells), IST1 (10 cells) or ALIX (11 cells) siRNAs treated cells were analyzed. A.U., arbitrary unit. **m** Seahorse FAO assays to examine the transport and oxidation of endogenous FAs in scrambled, Eto (40 μM) pretreated, VPS13D or TSG101 siRNAs treated HepG2 cells. The p values were labeled in front of the corresponding data points for Eto-pretreated (red), VPS13D siRNA (green), and TSG101 siRNAs (magenta) treated groups, compared with the scrambled siRNA treated group (blue). Oligo: oligomycin; FCCP: Carbonyl cyanide-p-trifluoromethoxyphenylhydrazone; RTN: rotenone; AA: Antimycin. Four independent assays were performed with similar results in (**i-l**), and at least three biological replicates were performed with similar results in **m**. The statistic analysis was performed with two-tailed unpaired student's t-test, and data are means ± SD in **i-m**. Scale bar, 10 μm in whole cell image and 2 μm in insets in **a–h**.

**The lipid transfer domain of VPS13D directly binds lipid fatty acid moieties in vitro.** VPS13D harbors a putative LTD at its N terminus, which is highly conserved among VPS13 paralogs and ATG2[27,52]. Two recent studies demonstrated that the LTD of fungi VPS13 proteins binds and transfers glycerophospholipids[17,53]. It is still unknown whether human VPS13D-LTD could directly binds and transfers lipid fatty acid moieties. To test whether VPS13D-LTD could directly bind lipids in vitro, we performed protein-lipid co-migration assays. Purified VPS13D-LTD co-migrated with nitrobenzoxadiazole (NBD)-labeled glycerophospholipids including phosphatidic acid (PA), phosphatidylserine (PS), phosphatidylethanolamine (PE), phosphatidylcholine (PC), and to a lesser extent, ceramide (Cer) (Fig. 7a), which was consistent with the findings of a previous study in which yeast VPS13α bound glycerophospholipids[17,53]. Besides phospholipids, purified VPS13D-LTD could bind NBD-labeled fatty acids C12 (NBD-C12) and BODIPY-labeled fatty acids C16 (BODIPY-C16) (Fig. 7a). The binding of purified VPS13D-LTD to FAs was stronger compared with that of VPS13D-LTD to phospholipids, as indicated by the ratio of FA fluorescence intensity to the amount of protein (Fig. 7b).

To further confirm that VPS13D binds FAs, we sought to estimate the stoichiometry of the binding of VPS13D-LTD to FAs via in vitro protein-lipids co-migration assays. Through titration, we found that one VPS13D-LTD molecule (320 residues) could bind ~5 molecules of BODIPY-C16 (Fig. 7c, d), and we further estimated that a full-length VPS13D molecule with a long groove serving as a lipid transfer channel at its N-terminus (~1500 residues) could bind ~25 molecules of BODIPY-C16, which agreed with the estimated stoichiometry of the binding of yeast VPS13 and ATG2 to phospholipids[17,27], and this result supported the notion that VPS13 is a bridge lipid transporter that is capable of binding tens of lipids at once[53]. In addition, to validate the finding that VPS13D binds endogenous, unlabeled FAs, we examined whether unlabeled FAs could outcompete BODIPY-C16 for binding to VPS13D-LTD. Competition assays indicated that palmitic acid, but not OA, could sufficiently outcompete BODIPY-C16 for binding to VPS13D-LTD in a dose-dependent manner (Fig. 7e, f). Collectively, these results supported the hypothesis that VPS13D-LTD directly binds FAs in vitro.

We further validated the role of the lipid transfer activities of VPS13D in FA transport in cells. Inspired by a recent study[27], in which a fragment of the ATG2A lipid transfer domain (mini-ATG2A, residues 1-345) could almost completely rescue the phenotype of ATG2 knockout lines upon overexpression, we tested whether VPS13D-LTD was indispensable for FA transfer from LD to mitochondria by rescuing the FA transfer defects induced by VPS13D suppression. Based on the high conservation of the LTDs between VPS13 and ATG2, we made a hydrophobic

surface mutant of VPS13D-LTD (L42R, L66R, L85R, I148R, I152E, I174R, M181R, F203E, L278R, L293R, W319R) (Supplementary Fig. 7d), which could still bind but not transport lipid fatty acids moieties[27].

VPS13D suppression substantially inhibited the Red C12 transfer from LDs to mitochondria, and this defect was significantly rescued by expression of siRNA-resistant full-length VPS13D^sfGFP (Fig. 7g–j). The wild-type VPS13D-LTD (VPS13D-LTD-WT) protein significantly rescued the VPS13D suppression-induced FAs trafficking defects after 5 h or 24 h of EBSS starvation (Fig. 7g, k). In contrast, the VPS13D-LTD mutant (VPS13D-LTD-mut) did not significantly rescue the FA trafficking defects (Fig. 7g, l). This line of evidence supported the hypothesis that VPS13D-LTD plays an important role in FAs trafficking at mitochondria-LD junctions. In addition, we found that VPS13D-LTP was substantially less effective in rescuing the FA trafficking defects than full-length VPS13D, suggesting that both the lipid binding/transfer activities and the LD remodeling activities of VPS13D-VAB and TSG101 are required for efficient FAs transfer from LDs to mitochondria.

## Discussion
Our results revealed a function of the disease-causing protein VPS13D at mitochondria–LD MCSs under starvation, and we proposed that VPS13D facilitated FA transfer at such MCSs through remodeling the morphology of the LD membranes via interaction with TSG101; and through binding FAs (Fig. 8).

Our findings suggest that VPS13D is a mitochondrial/LDs protein that is present at mitochondria−LD MCSs under starvation. However, it is possible that the localization of VPS13D may be subject to regulation by metabolic states or the cell cycle, as yeast VPS13 was found to localize to different organelle MCSs via different protein adaptors under different metabolic conditions[28,30]. In addition, adaptors that are responsible for recruiting VPS13D to the OMM are still unknown. This fundamental question awaits further investigation.

ESCRTs are considered to function in inverse budding reactions, in which ESCRT-III proteins form filaments that assemble on membranes with a negative curvature[33]. However, recent studies revealed that ESCRTs could also mediate classical-topology membrane remodeling, in which the ESCRTs form a double-stranded helical polymer that coats positively curved membranes[34]. For instance, IST1 promotes fission of endosomal tubules via recruitment of the microtubule-severing ATPase Spastin to mediate endocytic recycling[36], and both IST1 and CHMP1B mediate the transfer of FAs from lipid droplets to peroxisomes, possibly by modifying the LD morphology[42]. Our results further supported a direct role of ESCRT complexes in FAs trafficking from LDs to mitochondria

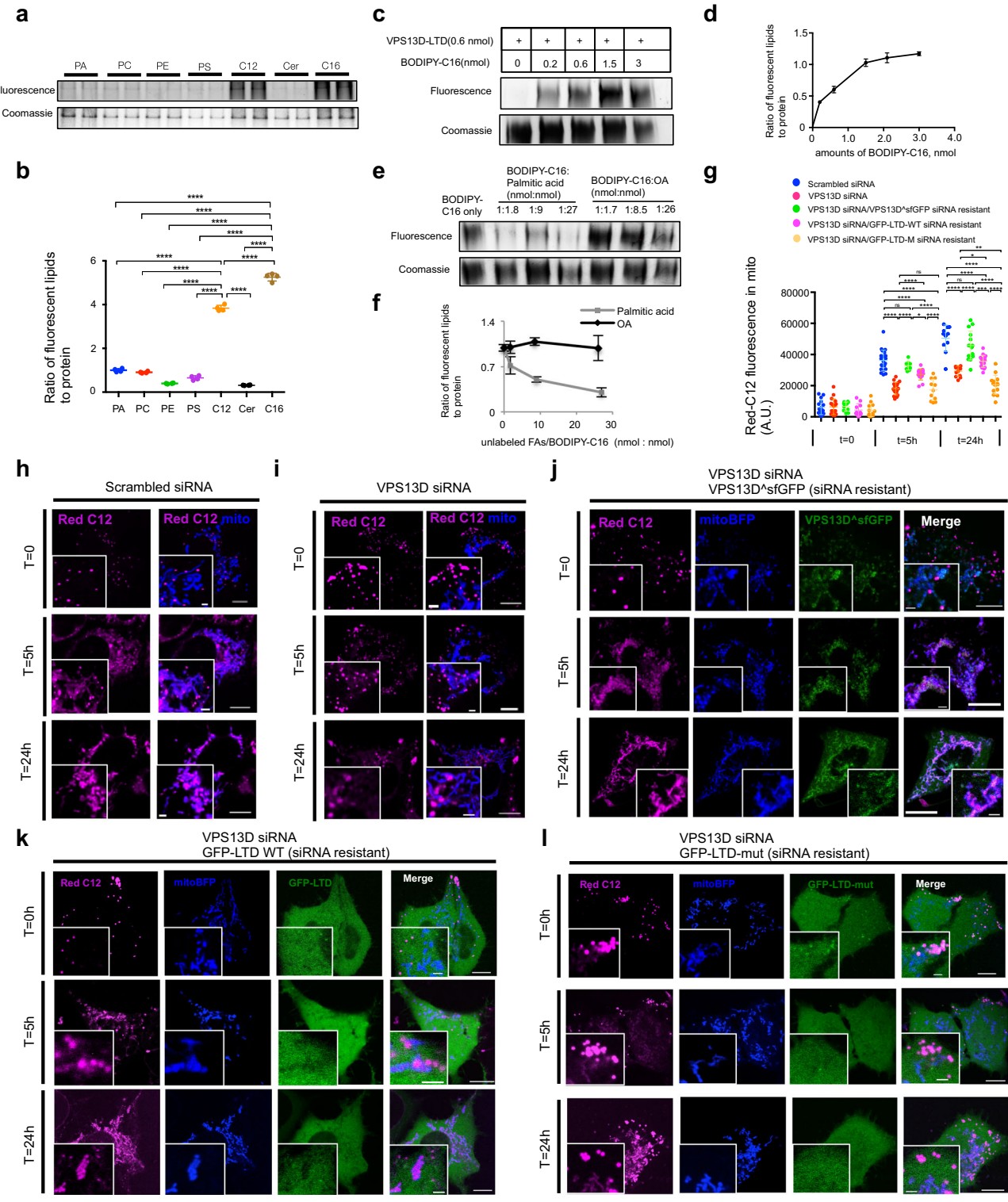

through the remodeling LD morphology. Because of the unique phospholipid monolayer of the LD membrane and the spatial segregation between triglycerides (inside LDs) and lipase (associated with the cytoplasmic face of the LD membrane), it is plausible that such remodeling of the LD membrane by VPS13D and TSG101 or other ESCRT components promotes the access of triglycerides to the membrane-associated lipase, and eventually facilitates triglyceride mobilization and subsequent FA trafficking.

A previous study showed that yeast Vps13α binds to glycerophospholipids but not to other lipid species in Expi 293 F cells by liquid chromatography–tandem mass spectrometry (LC/MS/MS)[17]. We postulated that two reasons to explain why free FAs were not found to co-purify with VPS13α in this previous work. First, the ionization efficiency of free FAs under electrospray ionization MS conditions without derivatization is poor, rendering the construction of a robust platform for profiling of various FAs in biological samples using LC-MS-MS very challenging[54]. In addition, the

**Fig. 7 VPS13D-LTD binds lipid fatty acids moieties in vitro. a** Purified VPS13D-LTD was incubated with fluorescent lipids and examined by native PAGE. Phospholipids and FAs, visualized by their fluorescence, comigrated with protein, visualized by coomassie blue staining. PA: phosphatidic acid; PS: phosphatidylserine; PE: phosphatidylethanolamine; PC: phosphatidylcholine; Cer: ceramide; C12: NBD-C12; C16: BODIPY-C16. **b** Fold change of lipid fluorescence to protein. Three biological replicates were performed with similar results. Ordinary one-way ANOVA (Tukey's multiple comparisons test). ****$p < 0.0001$. Mean ± SD. **c** Estimation of the stoichiometry of VPS13D-LTD binding to BODIPY-C16. **d** Fold change of lipid fluorescence to protein in **c**. Three biological replicates were performed with similar results. Mean ± SD. **e** Competition assays showing that unlabeled, non-BSA-conjugated palmitic acid, but not OA, sufficiently outcompeted BODIPY-C16 for binding to VPS13D-LTD. **f** Fold change of lipid fluorescence to protein in **e**. Three biological replicates were performed with similar results. Mean ± SD. **g** Quantification of Red C12 fluorescence in mitochondria upon 5 h of starvation. For $t = 0$: cells treated with scrambled ($n = 16$), VPS13D ($n = 18$), VPS13D siRNA/siRNA resistant VPS13D^sfGFP ($n = 11$), VPS13D siRNA/siRNA resistant GFP-LTD-WT ($n = 11$), VPS13D siRNA/siRNA resistant GFP-LTD-mut ($n = 12$) were analyzed. For $t = 5$ h: cells treated with scrambled ($n = 21$), VPS13D ($n = 18$), VPS13D siRNA/siRNA resistant VPS13D^sfGFP ($n = 10$), VPS13D siRNA/siRNA resistant GFP-LTD-WT ($n = 14$), VPS13D siRNA/siRNA resistant GFP-LTD-mut ($n = 11$) were analyzed. For $t = 24$ h: scrambled ($n = 11$), VPS13D ($n = 13$), VPS13D siRNA/siRNA resistant VPS13D^sfGFP ($n = 15$), VPS13D siRNA/siRNA resistant GFP-LTD-WT ($n = 18$), VPS13D siRNA/siRNA resistant GFP-LTD-mut ($n = 15$) were analyzed. A.U., arbitrary unit. Three independent assays were performed with similar results. Ordinary one-way ANOVA (Tukey's multiple comparisons test) within five groups in category of $t = 5$ h or $t = 24$ h. ****$p < 0.0001$; ***$p < 0.001$; **$p < 0.01$; *$p < 0.05$; ns: $p > 0.05$. Mean ± SD. **h–l** Red C12 pulse-chase assay ($t = 5$ h) of HEK293 cells treated with scrambled (**h**), VPS13D siRNAs (**i**), VPS13D siRNA/siRNA-resistant VPS13D^sfGFP (**j**), VPS13D siRNA/siRNA-resistant GFP-VPS13D-LTD-WT (**k**) or VPS13D siRNA/siRNA-resistant GFP-VPS13D-LTD-mut (**l**). Scale bar, 10 μm in whole cell image and 2 μm in insets in **h–l**.

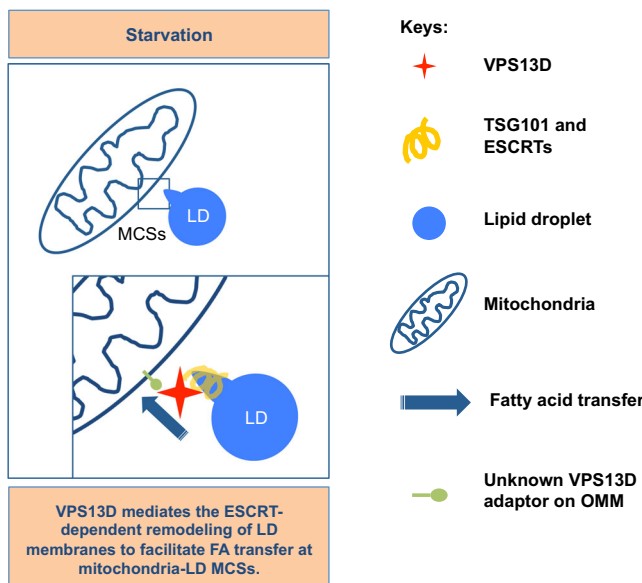

**Fig. 8 A proposed model of VPS13D and ESCRTs function at mitochondria-LD MCSs.** Upon starvation, VPS13D is accumulated at mitochondria-LD MCSs and mediates the ESCRT-dependent remodeling of LD membranes to facilitate FA transfer at these contacts. ESCRT, endosomal complexes required for transport; LD, lipid droplet; MCSs, membrane contact sites; OMM, outer mitochondrial membrane.

identification of free FAs is also easily affected by matrix effects and the sensitivity of detection is low[55]. Second, the release of FAs from triglycerides by lipases exclusively occurs upon stimulation by, for instance, starvation or exercise. Consistent with this, our results showed that barely any Red C12 was mobilized and transferred from LDs to mitochondria in complete medium (Fig. 7h, $t = 0$). Therefore, we postulated that the low abundance of FAs in cells cultured in complete medium might be another reason.

In addition, it should be noted that the results from the seahorse assays only indicated that VPS13D depletion modestly reduced the oxidation of FAs (Fig. 6m), but did not indicate a direct role of VPS13D in the transfer of FAs. Although our results demonstrated that the LTD of VPS13D bound FAs in vitro and was required for the efficient transfer of Red-C12 to mitochondria in vivo, more direct evidence is required to confirm that VPS13D directly transports endogenous FAs at MCSs.

## Methods

**Cell culture, transfection and stable line.** Human embryonic kidney 293 cells (ThermoFisher, R70507), African green monkey kidney fibroblast-like COS7 cell line (CRL-1651; ATCC), human liver cancer HepG2 cell line (HB-8065; ATCC) and Human embryonic kidney 293 T cells (HEK293T/17; CRL-11268; ATCC) were grown in Dulbecco's modified Eagle's medium (Invitrogen Life Technologies) supplemented with 10% fetal bovine serum (Gibco) and 1% penicillin/streptomycin. All of the cell lines used in this study are free of mycoplasma contamination. For transfection, cells were seeded at $4 \times 10^5$ cells per well in a six-well dish ~16 h before transfection. Plasmid transfections were performed in OPTI-MEM (Invitrogen Life Technologies) with 2 μl Lipofectamine 2000 per well for 6 h, followed by trypsinization and replating onto glass-bottom confocal dishes at ~3.5 × $10^5$ cells per well. Cells were imaged in live-cell medium (DMEM with 10%FBS and 20 mM Hepes no P/S) ~16–24 h after transfection. For all transfection experiments in this study, the following amounts of DNA were used per 3.5 cm well (individually or combined for cotransfection): 1 μg for VPS13D^sfGFP or siRNA-resistant VPS13D^sfGFP; 500 ng for truncated VPS13D mutations; 100 ng for Halo-TSG101; 500 ng for mito-BFP; For siRNA transfections, cells were plated on 3.5 cm dishes with 30–40% density, and 2 μl lipofectamine RNAimax (Invitrogen) and 50 ng siRNA were used per well. 48 h after transfection, second round of transfection was performed with 50 ng siRNAs. Cells were analyzed 24 h post-second transfection for suppression. All DNA plasmids used in this work are listed in Supplementary Table 1. DNA primers used for generating constructs and for quantitative PCR assays reported in this work are listed in Supplementary Table 2. Oligonucleotides for siRNA reported in this work are listed in Supplementary Table 3.

**Generation of stable pLVX-EF1α-VPS13D_N-GFP, pLVX-EF1α-VAB-GFP COS7 cell lines and stable Dox-inducible mitochondria-LD MCSs reporter HEK293 cells.** Day 1: HEK 293 T cells were seeded into 10 cm dishes with $3 \times 10^6$ cells 18 h before transfection. Day 2: transfections were performed with 4 μg pLVX-EF1α-VPS13D_N –GFP or pLVX-EF1α-VAB -GFP, 3 μg psPAX2, 2 μg pMD2.G; Day 3: COS7 cells were seeded to 10 cm dishes with $2 \times 10^6$ cells. Day 4: the supernatant from transfected HEK293T cells were collected 48 h post initial transfection, followed by filtering through a syringe with 0.22 μm filters. 1 ml of filtered HEK293T supernatant was added to COS7 cells. Day 5: COS7 cells were grown in puromycin-containing (1 μg/ml) medium, followed by verification by imaging and western blots. Procedure of generation of stable Dox-inducible mitochondria-LD MCSs reporter HEK293 cell line was similar to that of stable pLVX-EF1α-VPS13D_N-GFP COS7 cell line except that virus-infected HEK293 cells were grown in Dox-free medium containing G418 (2 μg/ml). For induction, Dox (0.1 μg/ml) was added to tetracycline-free medium and was verified by imaging and flow cytometry after 48 h.

**Antibodies and reagents.** Anti-VPS13D (A304-691A, Bethy Laboratories. Inc), Anti-GFP (AE011, Abclonal), anti-Halo (G9211; Promega), anti-Tubulin (100109-MM05T; Sinobiological), anti-actin (20536-1-AP; Proteintech), anti-VDAC1 (55259-1-AP; Proteintech), anti-TSG101 (A1692; Abclonal) were used at 1:1000 dilutions for western blots. Anti-VPS13D (A304-691A, Bethy Laboratories. Inc) and anti-TSG101 (SC7964; Santa Cruz Biotechnology) antibodies were used 1:100 for immunofluorescence. The following reagents were used in this study: Oleic acid (O1008; sigma); Palmitic acid (P0500; sigma); Doxycycline (1225984; Sigma), BODIPY 493/503 (ThermoFisher Scientific, D3922), BODIPY 558/568 (Thermo-Fisher Scientific, D2219), NBD-C12 (Abcam, Ab145361), BODIPY 558/568 C12 (ThermoFisher Scientific, D3835), BODIPY FL C16 (ThermoFisher Scientific,

D3821), Janilia Fluo® 646 HaloTag® Ligand (GA1120; Promega) and antibiotics such as G418 (10131027) and puromycin (A1113803) were obtained from ThermoFisher Scientific. All EM reagents were purchased from Electron Microscopy Sciences. Lipids were purchased from Avanti Polar Lipids: NBD-PC (810133), NBD-PE (810144), NBD-PS (810198), NBD-PA (810138), NBD-ceramide (810211).

**BODIPY 493/503, BODIPY 558/568 and BODIPY 558/568 C12 (Red C12) staining in live cell**. Cells were washed once with PBS, and were changed to complete medium containing 5 μM BODIPY493/503, 3 μM BODIPY558/568 or 1 μM BODIPY 558/568 C12 and incubated at 37 °C for indicated period of time. Cells were washed with PBS three times and changed to imaging medium (DMEM supplemented with 10% FBS and 20 mM Hepes without phenol red) prior to imaging.

**Halo staining in live cell**. Cells were incubated with complete medium with 5 nM Janilia Fluo® 646 HaloTag® Ligand for 30 min. Cells were washed three times with complete medium to remove extra ligands, followed by incubation for another 30 min. Medium was replaced with imaging medium to remove unconjugated Halo ligands that has diffused out of the cells prior to imaging.

**Live imaging by confocal microscopy**. Cells were grown on glass-bottom confocal dishes. Confocal dishes were loaded to a laser scanning confocal microscope (LSM780, Zeiss, Germany) equipped with multiple excitation lasers (405-nm, 458-nm, 488-nm, 514-nm, 561-nm, 633-nm) and spectral fluorescence GaAsP array detector. Cells were imaged with the 63 × 1.4-NA iPlan-Apochromat 63 x oil objective using the 405-nm laser for BFP, 488-nm for GFP, 561-nm for mStrawberry, OFP, tagRFP or mCherry. Cells were imaged in live cell chamber supplied with 5% CO$_2$ at 37 °C.

**Live cell imaging by Leica SP8 equipped with lightning super-resolution module**. Cells on confocal dishes were loaded to Leica SP8 equipped with lightning super-resolution module equipped with HC PL APO CS2 100 x / 1.4 oil objective, four lasers (415-nm, 499-nm, 567-nm, and 662-nm) with corresponding filters. Cells were imaged in OPTI-MEM with Hepes buffer in live cell chamber supplied with 5% CO$_2$ at 37 °C.

**Quantitative RT-PCR for detecting mRNA level of VPS13D and VPS13A**. HEK293 cells were transfected with either scrambled, VPS13D or VPS13A siRNAs. 4 days after transfection, RNA was isolated with Trizol (ThermoFisher Scientific) according to the instructions from the manufacturer. cDNA was reverse transcribed using RverTra Ace (TRT-101, TOYOBO) according to the directions of the manufacturer. The cDNA was analyzed using quantitative PCR with qPCR Mix (QPS-201, TOYOBO). The DNA primers for quantitative PCR assays are listed in Supplementary Table 2.

**Immunofluorescence staining**. Cells were fixed with 4% paraformaldehyde (50-259-99; Electron Microscopy Sciences) in PBS for 10 min at room temperature. After washing with PBS three times, cells were permeabilized with 0.1% Triton X-100 in PBS for 15 min on ice. Cells were then washed three times with PBS, blocked with 0.5% BSA in PBS for 1 h, incubated with primary antibodies in diluted blocking buffer overnight, and washed with PBS three times. Secondary antibodies were applied for 1 h at room temperature. After washing with PBS three times, samples were mounted on Vectashield (H-1000; Vector Laboratories).

**Antibody pre-clearing**. For the IF of endogenous VPS13D, anti-VPS13D antibody was pre-cleared. VPS13D depleted HEK293 cells were fixed in 4% paraformaldehyde for 15 min, rinsed with PBS, quenched in 50 mM NH4Cl in PBS, washed twice with PBS, permeabilized with PBS containing 0.1% Triton X-100 (PBX) for 10 min, and scraped in PBX with 1% Triton X-100. Antibodies were added to the fixed cells at the final concentration used for IF (1:100) and mixed by rotation overnight at 4 °C. The mixture was then centrifuged at 17,000 ×g for 20 min at 4 °C, and the supernatant containing cleared antibodies was used for IF.

**Differential centrifugation**. Cells were harvested from 2 × 10 cm$^2$ dishes at 90% confluence. The following steps were conducted at 4 °C or on ice. Cells were washed with pre-cold PBS once and homogenized in isolation buffer by ultrasonic. The homogenate was centrifuged at 1000g for 10 min to remove nuclei and unbroken cells. The resulting supernatant was centrifuged at 3500g for 10 min to obtain crude mitochondria. The resulting supernatant was further centrifuged at 12,000g for 10 min to collect the remaining crude mitochondria fractions. Western blots were performed using rabbit anti-VDAC1 (1:1000, A0810, Abclonal), rabbit anti-Tubulin (1:1000, 100109-MM05T, Sinobiological), and rabbit anti-GFP (1:2000, AE011, Abclonal) antibodies.

**Electron microscopy**. OA/EBSS-stimulated HEK293 cells, either expressing empty vectors or Halo-TSG101 and VAB-GFP were fixed with 2.5% glutaraldehyde in

0.1 M Phosphate buffer, pH 7.4 for 2 h at room temperature. After washing three times with 0.1 M Phosphate buffer, cells were scraped and collected with 0.1 M PBS followed by centrifugation at 1,100 ×g. The pellet was resuspended in PBS (0.1 M), and centrifuged at 1100g for 10 min. This step was repeated three times. The samples were post-fixed with pre-cold 1% OsO$_4$ in 0.1 M PBS for 23 h at 4 °C, followed by rinsing with PBS for three times (3 × 20 min). The samples were dehydrated in graded ethanol (50%, 70%, 85%, 90%, 95%, 2×100%) for 15 min in each condition. The penetrations were performed in an order of acetone-epoxy (2:1); acetone-epoxy (1:1); epoxy. Each round of penetrations was performed at 37 °C for 12 h. The samples were embedded in epoxy resin using standard protocols[56]. Sections parallel to cellular monolayers were obtained using a Leica EM UC7 with the thickness of 60–100 nm and examined under HT7800/HT7700. Mitochondria and LDs were identified based on their respective morphologies.

**GFP-trap assay**. GFP-Trap (GFP-trap agorose beads; GTA-100; ChromoTek) was used for detection of protein-protein interactions and the GFP-Trap assays were performed according to the manufacturer's protocol. 5% input was used in GFP traps unless otherwise indicated.

**Pulse-chase assay of BODIPY 558/568 C12 (Red C12)**. HEK293 cells were pulsed by an incubation in complete medium (DMEM with 10% fetal bovine serum) containing 1 μM BODIPY 558/568 C12 (ThermoFisher Scientific, D3835) for 16 h. Cells were washed with PBS three times and then chased in EBSS for the time indicated.

**Protein expression and purification**. pET28a-6xHis-SUMO-VPS13D-LTD (VPS13D residues 1–320) was expressed in TSsetta (DE3) Chemically competent cells (TSC04; Tsingke). Cells were grown at 37 °C to an OD$_{600}$ of 0.6–0.8, and then protein expression was induced by the addition of 0.1 mM Isopropyl β- d-1-thiogalactopyranoside. Cells were cultured at 16 °C for another 16 h. Cells were pelleted, resuspended in buffer A (20 mM Tris-Hcl, pH 8.0, 300 mM NaCl, 10 mM imidazole, 1 mM DTT and 10% glycerol) supplemented with protease inhibitor cocktail (C0001; TargetMoI) and lysed in a JY88-IIN cell disruptor. Cell lysates were centrifuged at 14,000g for 30 min. Supernatant was incubated with Ni-NTA resin (30230; QIAGEN) for 2 h at 4 °C, and then the resins were passed through via gravity flow. The Resins were washed with 5-bed volumes of buffer B (20 mM Tris-Hcl, pH 8.0, 300 mM NaCl, 20 mM imidazole, 1× protease inhibitor cocktail (C0001; TargetMoI)), followed by 5-bed volumes of buffer C (20 mM Tris-Hcl, pH 8.0, 300 mM NaCl, 50 mM imidazole, 1× protease inhibitor cocktail (C0001; TargetMoI)) and 5-bed volumes of buffer D (20 mM Tris-Hcl, pH 8.0, 300 mM NaCl, 100 mM imidazole, 1× protease inhibitor cocktail (C0001; TargetMoI)). Retained proteins were eluted from the resin with buffer B supplemented either with 200 mM imidazole or 500 mM imidazole, respectively. The eluted protein solutions were concentrated in a 3-kDa molecular weight cutoff Amicon centrifugal filtration device (UFC5003).

**In vitro lipid-binding assay**. 19 μl purified VPS13D LTD (1.5 mg/ml) was mixed with 1 μl of either NBD-labeled PA, PC, PE, PS, ceramide, C12 (1 mg/ml in methanol) or BODIPY-C16 (1 mg/ml in DMSO) in 20-μl total reaction volumes and incubated at 4 °C for 3 h. Samples were loaded onto 12% Precast Native gels with Hepes-Tris buffer and run for 4 h at 90 V on ice. NBD/BODIPY fluorescence was visualized using a Bio-rad ChemiDoc XRS + (170-8265). Then gels were stained with Coomassie blue G250 (20279; ThermoFisher Scientific) to visualize total proteins.

**Oxygen consumption rate (OCR) measurements using Seahorse XF Cell Mito Stress Test**. HepG2 cells were seeded in XF24 Cell Culture Microplate in normal growth medium. 24 h prior to the assay, growth medium was replaced with substrate-limited medium containing 0.5 mM glucose, 1 mM GlutaMAX, 0.5 mM carnitine, and 1% FBS. 45 min prior to the assay, the cells were washed for two times with FAO Assay Medium containing 111 mM NaCl, 4.7 mM KCl, 1.25 mM CaCl$_2$, 2 mM MgSO$_4$, 1.2 mM NaH$_2$PO$_4$, 2.5 mM glucose, 0.5 mM carnitine, and 5 mM HEPES adjusted to pH 7.4 at 37 °C. 375 μL/well FAO assay medium was added to the cells and cells were further incubated in a non-CO$_2$ incubator for 30 min at 37 °C. Meanwhile, the assay cartridge was loaded with XF Cell Mito Stress Test compounds (final concentrations: 2.5 μg/mL oligomycin, 2 μM FCCP, 2 μM rotenone, and 4 μM antimycin A). 15 min prior to starting the assay, cells were incubated in Eto-containing FAO Assay Medium (final concentration: 40 μM) to allow for efficient inhibition of CPT1. After 15 min incubation at 37 °C in a non-CO2 incubator, the XF Cell Culture Microplate was inserted into the XF24 Analyzer and run the XF Cell Mito Stress Test according to the manufacturer's instructions.

**Image analysis**. All image analysis and processing was performed using imageJ (2.0.0-rc-66/1.52b; National Institutes of Health). Colocalization-based analysis of mitochondria-LDs MCSs was performed using a plugin named colocalization in imageJ with the following settings: Ratio (0-100%): 50; Threshold channel 1(0-255): 50; Threshold channel 2 (0-255): 50; Display value (0-255): 255. MCSs were

automatically identified by colocalization plugin with white pixels representing potential MCSs. For the measurement of mitochondrial Red C12 fluorescence intensity, an area encompassing the whole mitochondrial network was manually selected, and was measured via imageJ. The Red C12 fluorescence intensity of the whole cell areas was also calculated via imageJ.

**Statistics and reproducibility.** All statistical analyses and p-value determinations were performed with GraphPad Prism6 and Microsoft Excel (2011, 2013). All the error bars represent mean ± SD. To determine $p$-values, ordinary one-way ANOVA with Tukey's multiple comparisons test were performed among multiple groups; and a two-tailed unpaired student $t$-test was performed between two groups. A $p$-value of <0.05 was considered statistically significant. All western blots, immunoprecipitation, seahorse assays and in vitro protein-lipid binding assays were from at least three biological replicates and representative results are shown. Images from indirect IF of endogenous VPS13D using anti-VPS13D antibodies in HEK293 cells are representative from four independent experiments. Images from indirect IF of endogenous TSG101 using anti-TSG101 antibodies in HEK293 cells are representative from three independent experiments. Live imaging of transiently transfected cells expressing full length VPS13D^sfGFP was performed in HEK293 cells as VPS13D^sfGFP was too large to efficiently transfect other cell lines. Live imaging of transiently transfected cells expressing VPS13D mutants was performed in two cell lines: COS7 and HEK293. Cells expressing VPS13D^sfGFP or VPS13D mutants were imaged between 5 and 40 independent live-cell imaging sessions; cells expressing Halo-TSG101 and TSG101 mutants were imaged between 3 and 25 different sessions. Cells expressing VAPB-GFP and Halo-TSG101 were imaged at least 30 independent live-cell imaging sessions and repeated in three biological replicates in TEM with similar results. The splitGFP-based mito-LD MCSs reporter cells were imaged in at least four independent live-cell imaging sessions, and were applied to flow cytometry three times under two different conditions of Dox inductions, respectively, with similar results. The ddGFP-based mito-LD MCSs reporter cells were imaged in three independent live-cell imaging sessions, and were applied to flow cytometry three times with similar results. HEK293 cells treated either with scrambled or VPS13D siRNAs were imaged between 3 and 10 independent live-cell imaging sessions for the quantifications of mitochondria-LD interactions. HEK293 cells treated either with scrambled, VPS13D, TSG101, CHMP4B, CHMP1B, CHMP6, IST1 or ALIX siRNAs were imaged in at least 4 independent live-cell imaging sessions. VPS13D-suppressed HEK293 cells rescued either with siRNA-resistant full length VPS13D^sfGFP, siRNA-resistant GFP-LTD or siRNA-resistant GFP-LTD Mutant were imaged in three independent live-cell imaging sessions.

**Reporting summary.** Further information on research design is available in the Nature Research Reporting Summary linked to this article.

## Data availability

All the data and relevant materials, including reagents and primers, that support the findings of this study are available from the corresponding author upon reasonable request. Source data are provided with this paper.

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

## Acknowledgements
We are grateful to Marianna Leonzino (Yale University School of Medicine) and Karin M. Reinisch (Yale University School of Medicine) for advices in transfections and expressions of VPS13 constructs in mammalian cells. We thank Lin Deng (Shenzhen Bay Institute) for discussions and reagents; we thank Guangjun Cai (Huazhong University of Science and Technology, HUST) and Ping Liu (the Optical Bioimaging Core Facility of WNLO-HUST) for imaging assistance. We thank Sheng Wu (Servicebio) for EM assistance. W.J. was supported by National Natural Science Foundation of China (91854109; 31701170), and the Program for HUST Academic Frontier Youth Team (2018QYTD11). J.X. was supported by National Natural Science Foundation of China (81901166).

## Author contributions
J.W., N.F., J.X., Y.D., and W.J. conceived the project and designed the experiments. J.W., N.F., J.X., Y.D. Y.C., and W.J. performed the experiments. J.W., N.F., J.X., Y.D. Y.C., and W.J. analyzed and interpreted the data. W.J. prepared the manuscript with inputs and approval from all authors.

## Competing interests
The authors declare no competing interests.
