## [Peer Review File · Nature Communications]

Reviewers' comments:

Reviewer #1 (Remarks to the Author):

In this paper, the authors investigate the localization and function of VPS13D at membrane contact sites. VPS13D is a member of the VPS13 family of presumed lipid transporters, which are implicated in several different diseases and have been the focus of recent investigations. While other members of this family have been localized to sites of contacts between the ER and mitochondria, endolysosomes or lipid droplets, the localization of VPS13D has been unclear. Here, the authors show that VPS13D is targeted to mitochondria-LD contact sites under specific nutrient conditions (ie. oleate loading followed by starvation). They identified Vps13D N and C-terminal fragments that when overexpressed were targeted to either mitochondria or LD in an oleate-dependent manner. A central "DUF1162" domain-containing fragment was found to bind ubiquitin and the ESCRT protein Tsg101 and all three partly colocalized at lipid droplets in oleate-loaded cells.

While I did not find experiments based on the overexpression of protein fragments entirely compelling (see below), knockdown of VPS13D reduced both mitochondrial-LD contacts and the localization of Tsg101 to LDs. Knockdown of VPS13D and several ESCRT subunits also inhibited the transport of fatty acids from LDs to mitochondria. This was an interesting result, even though the reason for the inhibition of FA trafficking was not clearly defined. VPS13 proteins have been shown to transport glycerophospholipids *in vitro*, rather than fatty acids, and the requirement for ESCRT proteins, which appear to be recruited by VPS13D, suggests that FA transport is likely downstream of VPS13D function.

Knockdown of 13D had a number of other effects: it altered the size/morphology and distribution of lipid droplets and mitochondria, reduced the frequency of mitochondrial-LD contacts, and affected mitochondrial DNA replication and calcium uptake. These functional effects, which impact both mitochondria and LDs, support a role for VPS13D at LD/mitochondria contacts. While other studies have shown a role for VPS13D in maintaining mitochondrial morphology and/or distribution, many of these functional impacts have not been previously described and may be highly relevant for understanding disease mechanisms.

One of the most striking effects of VPS13D knockdown was the elaboration of ER-mitochondrial contacts. The idea of cross talk between different types of contact sites is often discussed in the literature, but the mechanistic basis of such cross talk is not well understood. The authors propose that loss of VPS13D increases the amount of VAPB, which in turn enhances ER-mitochondrial tethering by binding PTPIP51. Knocking down both VAPB and PTPIP51 rescued the hypertethering defect. The authors further suggest that p97 provides a link between VPS13D and VAPB. Inhibiting p97 also caused hypertethering, and while it was suggested that p97 acts by reducing VAPB levels, this was not directly examined, nor was VAPB overexpression shown to cause the hypertethering phenotype. How VPS13D interacts with VAPB and p97 to regulate the extent of tethering remains unclear. P97 was previously shown to degrade aggregated mutant forms of VAPB, but to my knowledge it has not been shown to regulate wild type VAPB levels. Taken together these observations are interesting and suggestive, but more work will need to be done to understand how this cross talk really works.

Overall, this manuscript presents a vast amount of data that covers a wide range of concepts and presents a large number of functional assays. The paper makes many interesting and often exciting observations. However, many of the findings are presented in the supplementary materials, which makes the paper difficult to read. Moreover, the authors often fail to thoroughly examine and test one set of observations/predictions before jumping to a new aspect of their study. The lack of focused in-depth analysis is particularly apparent in the discussion, which for the most part fails to discuss their findings in the context of previously published work, and instead largely poses questions for future study. A better-written discussion would also tie together the very large number of different experiments and better highlight the impact of the work.

Specific comments:

1. In general, I did not find the experiments with overexpressed protein fragments very compelling. Several different fragments (N-terminal, DUF1162, C-terminal mutants) were localized to mitochondria under different nutrient conditions, but misfolded proteins are known to associate with the outer mitochondrial membrane (Nature 2017 543:443; EMBO J 2018 38:e98786). It's also not uncommon for overexpressed proteins normally present at other organelles to be found at lipid droplets, especially under conditions of oleate loading. Mutating the mitochondrial and LD targeting signals in the context of full-length VPS13D would add confidence that these are bona fide localization signals.
2. I did not find the experiments present in FigS4 (ie binding to FFAT motifs) helpful. These are negative results: FFAT motifs are difficult to predict and it is not clear that small fragments containing these putative motifs would be sufficient to target organelles. The GFP TRAP data suggest VAPB does bind VPS13D, which could mean that the wrong FFAT motifs were examined, multiple FFATs are required for binding, or that binding is FFAT-independent (which can be tested by examining binding of VPS13D to VAPB mutants lacking the FFAT binding site). If these data are to be included it would be enough to present just the table in S4B, the examples are not needed.
3. Because Fig2E is missing an important GFP-only control we can't know if the binding to the UEV domain is above background levels.
4. Knockdown of 13D KD increases the level of VAPB, but not PTPIP51, as shown by western. P97 is hypothesized to also control VAPB level, yet this has been shown for mutant, aggregation-prone forms of VAPB and not wild type VAPB, to my knowledge. Does p97 inhibition also increase levels of VAPB or VAPA? Does overexpressing VAPB cause hyper-tethering as predicted?
5. The authors conclude that the DUF1162 domain bound K63 ubiquitin and interacted with TSG101. However, the PSAP motif that is important for this interaction does not lie within the computationally-defined DUF1162 domain, but is instead in an upstream region. A recent paper showed this upstream region was part of a larger repeat structure that includes the smaller, annotated DUF1162 domain. Therefore, it is not appropriate to use the term DUF1162 here. Two recent papers have referred to the region containing the entire repeat structure, alternatively, as the "VAB domain" or "WD40 repeat" region. One of these terms should be used instead of "DUF1162".
6. The authors found that two independent regions of Tsg101 (ie. UEV domain or PR-CC-SB domains) bound the DUF1162 domain. References were cited to suggest this bipartite binding has been observed previously, but in both examples given, a single domain of tsg101 (the UEV domain) bound two different regions of Hrs or Tal. This should be clarified in the text.
7. The authors state: "For instance, yeast VPS13 contributed to the growth of the prospore membrane (58) (59) by associating with spindle pole protein Cdc31 (60)". This is not true: no direct link between the VPS13-Cdc31 interaction and prospore membrane growth has been demonstrated.
8. The discussion fails to discuss the current findings in the context of the relevant literature. Specifically, I would expect to see much more discussion of Anding et al 2018 (Current Biology 2018. 28:287). This paper identified VPS13D, Tsg101 and the ESCRTII component VPS36 in a genetic screen for mitophagy in drosophila, identified VPS13D as a ubiquitin-binding protein (via the UBA domain) and showed that reducing mitochondrial fusion rescued defects caused by loss of VPS13D. These results are similar to the findings presented in this paper yet have many key differences.
9. In general, I found the figures hard to follow because so much of the data is presented in supplementary figures, the text does not always refer to the correct figure, panels are not always

presented in order, and figure legends give few details (eg. don't always explain the markers used, don't always have a heading stating the key observation).

Specific errors in figures include:

-Figure 2I, the second set of panels has "halo-VPS23" as a label. Is this a typo? Tsg101 is generally considered to be "mammalian VPS23"

-Figure S11H: in label for y-axis, what are GFP-EMCs? Any new abbreviations should be described in legend.

-Figures S12 and S13 appear to be swapped.

-The text says "our GFP-Trap assay using HEK293 cells...identified...VAPB-PTPIP51 as potential VPS13D interactors (Fig. S7D)" yet S7D does not show GFP-Trap data.

-In Fig4I, the spacing of the labels on the figure itself makes it hard to see which are mitochondrial vs ER

-in Fig4H, the inset scale is too small to read.

- In FigS8, I can't distinguish red vs pink in merged images. Pick different colors, or show each color before merge? A kymograph might be a better way to look at this data

- several figures (eg. 2F, S10D) use a t-test instead of ANOVA with post-hoc test, which would be more appropriate.

Reviewer #2 (Remarks to the Author):

This study shows that over expressed VPS13D localizes to contact sites between lipid droplets. It identifies domains in VPS13D required for it to interact with both organelles. It identifies a domain VPS13D that interacts with the ESCRT component TGS101, which is required for VPS13D localization. It also presents evidence that contacts between lipid droplets and mitochondria are reduced when VPS13D is knocked down and that VPS13D is necessary for fatty acid transport to mitochondria. While these results are interesting, evidence that VPS13D plays a significant role in maintaining lipid droplet-mitochondria contacts and fatty acid transport is not well supported. There are number of concerns.

1. Lipid transport is assessed using BODIPY-labeled fatty acids. There are important concerns about whether the transport and metabolism of these fatty acids is similar to that of endogenous, unlabeled fatty acids. It is necessary to show that the movement and metabolism BODIPY-labeled fatty acids is similar to that of endogenous fatty acids. In addition, while the abstract suggests that the movement of fatty acids from lipid droplets to mitochondria is required for fatty acid oxidation, rates of oxidation have not been assessed. It has also not been shown that the BODIPY fatty acids are oxidized or even conjugated to CoA or carnitine, which is required for the uptake and oxidation of endogenous fatty acids by mitochondria. The rate of fatty acid transport should be directly assessed, and the results validated with endogenous fatty acids. Finally, there is no in vitro evidence that VPS13D can directly bind and transport fatty acids.

2. The images in Fig. S9B suggest that there are mitochondria that are not in contact with lipid droplets. Is the rate of fatty acid transport to these mitochondria less than those in contact with lipid

droplets?

3. The localization of VPS13D at lipid droplet-mitochondria junctions is only seen with overexpressed protein. The paper would be stronger if could be shown that endogenous VPS13D is also at these contact sites. I realize this is a challenging problem because VPS13D is a low abundance protein.

4. The evidence that knockdown of VPS13D reduces lipid droplet-mitochondria contacts is not strong. Using split-GFP is not a good method to assess contacts since, as the authors point out, the interaction is irreversible. No controls are included to show that the proteins are expressed at the same level in the different cell lines. The images in Fig. S9 suggest that most mitochondria remain in contact with lipid droplets after VPS13D knockdown. There are fewer contacts per mitochondria length but also, it seems, fewer lipid droplets. There is no evidence that fewer contacts reduces lipid transport rates.

5. While the results in Figure 4 are interesting, it is not clear that they add to the central point of this study.

6. It is not clear whether the authors are proposing that the ESCRT proteins are part of a tether or whether play a direct role in fatty acid transport. The second is a more interesting idea, but there is no evidence to support it.

Reviewer #3 (Remarks to the Author):

This is a paper that presents interesting results that will be of sufficient interest and novelty for consideration in Nature Comms. A potential new function for ESCRT complex components. The problem for me is the conveyance of the message - it is a sprawling, disengaging read. For my mind there is way too much supplementary material- (16 figures) vs 4 in the main text. Even the summary model is a supplementary figure. I think the authors should consider a major overhaul of the manuscript to make it more direct and succinct- concentrating on the firmest data. This does not include their ubiquitin binding data which are particularly opaque and rely on constructs that have had multiple lysines mutated, which is not clearly explained. The paper is largely descriptive and one is left wondering how the observed phenomena relate to the near universal property of ESCRT complexes in constriction and severance of membranes. Error bars are not always clearly explained and some data handling is a bit unclear. For example data presented in Fig 1C and D , cell counts are given, but does this amalgate Biological repeats?

Point-by-point response to Reviewer's Comments

Reviewer #1 (Remarks to the Author):

In this paper, the authors investigate the localization and function of VPS13D at membrane contact sites. VPS13D is a member of the VPS13 family of presumed lipid transporters, which are implicated in several different diseases and have been the focus of recent investigations. While other members of this family have been localized to sites of contacts between the ER and mitochondria, endolysosomes or lipid droplets, the localization of VPS13D has been unclear. Here, the authors show that VPS13D is targeted to mitochondria-LD contact sites under specific nutrient conditions (i.e. oleate loading followed by starvation). They identified Vps13D N and C-terminal fragments that when overexpressed were targeted to either mitochondria or LD in an oleate-dependent manner. A central "DUF1162" domain-containing fragment was found to bind ubiquitin and the ESCRT protein Tsg101 and all three partly colocalized at lipid droplets in oleate-loaded cells.

We thank the reviewer for the positive comments.

While I did not find experiments based on the overexpression of protein fragments entirely compelling (see below), knockdown of VPS13D reduced both mitochondrial-LD contacts and the localization of Tsg101 to LDs. Knockdown of VPS13D and several ESCRT subunits also inhibited the transport of fatty acids from LDs to mitochondria. This was an interesting result, even though the reason for the inhibition of FA trafficking was not clearly defined. VPS13 proteins have been shown to transport glycerophospholipids *in vitro*, rather than fatty acids, and the requirement for ESCRT proteins, which appear to be recruited by VPS13D, suggests that FA transport is likely downstream of VPS13D function.

Thank you for your insightful comments. In accordance with the reviewer's concerns, we have provided the following data on full length VPS13D (please see specific comment 1) in our revised manuscript.

To further identify the potential role of VPS13D in FA transport, we carried out a series of study. VPS13D harbored a lipid transfer domain (LTD) at its N terminus, which was conserved to yeast VPS13 (De et al. (2017) JCB) and ATG2 (Valverde et al. (2018) JCB). As you have mentioned, two recent studies demonstrated that fungi VPS13 lipid transfer domain bound and transferred glycerophospholipids (Kumar et al. (2018) JCB; Li et al. (2020) JCB). To test whether VPS13D-LTD could directly

bind lipid fatty acids moieties *in vitro*, we performed protein-lipids co-migration assays. We found that, besides phospholipids, purified VPS13D-LTD could bind both medium-chain fatty acids C12 and long-chain fatty acid C16 (Fig.7A&B). Our work supported that VPS13D-LTP could directly bind FAs *in vitro*.

We further validated the role of the lipid transfer activities of VPS13D in FA transport in cells. Inspired by a recent study by Valverde et al (2019), in which a fragment of lipid transfer domain of ATG2A (mini-ATG2A, 1-345aa) can near-complete rescue the phenotype of ATG2 knockout upon overexpression (at level around 10 times endogenous), we tested whether VPS13D-LTD was sufficient for FA transfer from LD to mitochondria by rescuing the FA transfer defects induced by VPS13D suppression. Based on the high conservation between VPS13 and ATG2 at their N terminal lipid transfer domain, we made a hydrophobic surface LTD mutant (VPS13D-LTD mutant: L42R , L66R , L85R , I148R , I152E , I174R , M181R , F203E , L278R, L293R , W319R) (Fig. 7C), which could still bind but could not transport lipid fatty acids moieties (Valverde et al. (2018) JCB; Li et al. (2020) JCB). As shown in Fig.7G, VPS13D-LTD substantially rescued the VPS13D suppression induced FAs trafficking defects upon 5 h or 24 h EBSS starvation. Instead the VPS13D-LTD mutant did not rescue the FAs trafficking defects (Fig.7H), as indicated by colocalization between red C12 and mitochondrial marker via the Pearson's correlation coefficient (Fig.7C). We also found that the rescue effects of VPS13D-LTP were substantially less than that of full-length protein, though the VPS13D-LTD was highly expressed by a CMV promoter. This result suggested that, besides the lipid transfer activities, the VPS13D-TSG101 interactions also played an important role in this process. In summary, our results supported an important role of VPS13D-LTD in FAs trafficking at mitochondria-LD MCSs.

The previous study by Kumar (2018) JCB showed that yeast Vps13 α bound to glycerophospholipids but not other lipid species in Expi 293F cells by liquid chromatography–tandem mass spectrometry (LC/MS/MS). We postulated that there were two reasons to explain why FAs were not identified with co-purified VPS13 α by LC/MS/MS in the previous work. First, the ionization efficiency of FAs under electrospray ionization mass spectrometry conditions is poor, rendering the construct of a robust platform for profiling of various FAs in biological samples using LC-MS-MS very challenging (Jun Mok et al (2016) RSC Advances). Besides, the identification of FFAs is also easily affected by matrix effects and the sensitivity is low (Narayana et al. (2015) Chemistry & Biology). Second, the release of FAs from triglyceride by lipases only occurred upon stimulation, for instance, starvation or

exercise. Consistently, our results showed that red C12 was barely mobilized and transferred from LDs to mitochondria in complete medium (Figs.7E-I). Therefore, we postulated that the low abundance of FAs in cells cultured in complete medium might be another reason

These results were shown below and incorporated into Fig. 7 in the revised manuscript.

Figure 7

One of the most striking effects of VPS13D knockdown was the elaboration of ER-mitochondrial contacts. The idea of cross talk between different types of contact sites is often discussed in the literature, but the mechanistic basis of such cross talk is not well understood. The authors propose that loss of VPS13D increases the amount of VAPB, which in turn enhances ER-mitochondrial tethering by binding PTPIP51. Knocking down both VAPB and PTPIP51 rescued the hypertethering defect. The authors further suggest that p97 provides a link between VPS13D and VAPB. Inhibiting p97 also caused hypertethering, and while it was suggested that p97 acts by reducing VAPB levels, this was not directly examined, nor was VAPB overexpression shown to cause the hypertethering phenotype. How VPS13D interacts with VAPB and p97 to regulate the extent of tethering remains unclear. P97 was previously shown to degrade aggregated mutant forms of VAPB, but to my knowledge it has not been shown to regulate wild type VAPB levels. Taken together these observations are interesting and suggestive, but more work will need to be done to understand how this cross talk really works.

Thank you for your valuable comments. As you suggested, we carefully examined the effects of p97 on WT/P56S/T46I VAPB and the effects of VAPB overexpression on ER-mitochondria interactions (please see specific comment 4).

Overall, this manuscript presents a vast amount of data that covers a wide range of concepts and presents a large number of functional assays. The paper makes many interesting and often exciting observations. However, many of the findings are presented in the supplementary materials, which makes the paper difficult to read. Moreover, the authors often fail to thoroughly examine and test one set of observations/predictions before jumping to a new aspect of their study. The lack of focused in-depth analysis is particularly apparent in the discussion, which for the most part fails to discuss their findings in the context of previously published work, and instead largely poses questions for future study. A better-written discussion would also tie together the very large number of different experiments and better highlight the impact of the work.

Thank you for your instructive suggestions, we do recognize the structural issues of the manuscript. According to your advice, we reorganized the manuscript, and tried to clarify the data and give more rational explanation. We also have rewritten the discussion section in the newly submitted manuscript to highlight the importance of our work.

Specific comments:

1. In general, I did not find the experiments with overexpressed protein fragments very compelling. Several different fragments (N-terminal, DUF1162, C-terminal mutants) were localized to mitochondria under different nutrient conditions, but misfolded proteins are known to associate with the outer mitochondrial membrane (Nature 2017 543:443; EMBO J 2018 38:e98786). It's also not uncommon for overexpressed proteins normally present at other organelles to be found at lipid droplets, especially under conditions of oleate loading. Mutating the mitochondrial and LD targeting signals in the context of full-length VPS13D would add confidence that these are bona fide localization signals.

Thank you for pointing out this problem. We agree that overexpression of protein fragments can be aggregation-prone and mistargeted to mitochondria or LD. To strengthen our results, we made VPS13D mutants with deletion of mitochondrial or LD localization signals and tested if these VPS13D mutants lost its membrane localizations by confocal microscopy. As shown in Fig.2G, the VPS13D mutant with three point mutations in two amphiphilic helices of VPS13_C domain (L3991Q L4052Q L4053Q) failed to associate with LDs but localized to out mitochondria membrane (OMM) (Fig.2G). We observed that the mitochondria appeared to be fragmented upon expression of this mutant (Fig.2G).

Next, we made another VPS13D mutant (GFP-VPS13D- Δ N), in which the N terminal region (1-1878 residues) containing VPS13D_N was deleted. We observed that the mutant did not localize to OMM but associated with LDs (Fig.3F). Disruption of the two amphipathic helices by introduction of the three point mutations (L3991Q L4052Q L4053Q) to VPS13D- Δ N blocked the its LD localization (Fig.3G).

These results were shown below and incorporated into Fig. 2G and Fig.3F&H in the revised manuscript.

Figure 2

G

Figure 3

F

G

2. I did not find the experiments present in FigS4 (i.e. binding to FFAT motifs) helpful. These are negative results: FFAT motifs are difficult to predict and it is not clear that small fragments containing these putative motifs would be sufficient to target organelles. The GFP TRAP data suggest VAPB does binds VPS13D, which could mean that the wrong FFAT motifs were examined, multiple FFATs are required for binding, or that binding is FFAT-independent (which can be tested by examining binding of VPS13D to VAPB mutants lacking the FFAT binding site). If these data are to be included it would be enough to present just the table in S4B, the examples

are not needed.

Thanks for your insightful advice. We agree with your concern about the accuracy of computational prediction of the FFAT motif. As you suggested, our results cannot preclude the possibility of VPS13D targeting ER through other mechanisms. Therefore, we have modified this section of results by deleting the image examples (original Figs.S4C-E), and we toned down this section of results by stating that ‘VPS13D truncated fragments containing the FFAT motifs predicted by computational algorithm listed in the table in Fig.S4B failed to associate with ER membrane. However, our results did not preclude the possibility that VPS13D associated with ER membrane through other unknown mechanisms.

3. Because Fig2E is missing an important GFP-only control we can't know if the binding to the UEV domain is above background levels.

We are sorry for missing this important control. We performed GFP-Trap assay between GFP tag-only and UEV domain or PR-CC-SB region of TSG101 in COS7 cells. We did not observe obvious interactions between GFP tag and Halo-UEV or Halo-PR-CC-SB in this assay.

These results were shown below and incorporated in Fig. S6D in our revised manuscript.

Figure S6

D

4. Knockdown of 13D KD increases the level of VAPB, but not PTPIP51, as shown

by western. P97 is hypothesized to also control VAPB level, yet this has been shown for mutant, aggregation-prone forms of VAPB and not wild type VAPB, to my knowledge. Does p97 inhibition also increase levels of VAPB or VAPA? Does overexpressing VAPB cause hyper-tethering as predicted?

Thanks for your insightful comment. As you suggested, we performed western blotting assays and showed overexpression of p97 substantially decreases the level of GFP-VAPB similar to that of GFP-VAPB P56S compared to non-p97 expressing group in U2OS cells (Fig. 9H&I).

Interestingly, we did not observe a decrease in the level of VAPB T46I upon p97 overexpression (Fig. 9H&I). We confirmed this interesting phenotype in other cell lines including HeLa, COS7 and HEK293 cells, indicating this VAPB mutant was resistant to p97-mediated protein degradation pathway.

These results were shown below and incorporated in Fig.9H&I in revised manuscript.

Figure 9

Figure S14

We also tested whether VAPB or VAPB/PTPIP51 overexpression increased ER-mitochondrial associations by confocal microscopy and flow cytometry in ER-mitochondrial MCSs reporter cells. Overexpression of VAPB substantially enhanced the interactions between ER and mitochondria either by microscopy (Fig. S15A-C) or flow cytometry (Fig. S15D-H), and co-expression of VAPB and PTPIP51 enhanced ER-mitochondrial associations to a larger extent compared to VAPB overexpressing cells (Fig. S15D-H). This finding is consistent with a previous work (Stoica et al. (2014) Nature Communications). Taken together, VAPB overexpressing could cause ER-mitochondrial hyper-tethering.

These results were shown below and incorporated in Fig. S14 in revised manuscript.

Figure S14

5. The authors conclude that the DUF1162 domain bound K63 ubiquitin and interacted with TSG101. However, the PSAP motif that is important for this interaction does not lie within the computationally-defined DUF1162 domain, but is instead in an upstream region. A recent paper showed this upstream region was part of a larger repeat structure that includes the smaller, annotated DUF1162 domain. Therefore, it is not appropriate to use the term DUF1162 here. Two recent papers have referred to the region containing the entire repeat structure, alternatively, as the “VAB domain” or “WD40 repeat” region. One of these terms should be used instead of “DUF1162”.

Thank you for the advice. We have changed the “DUF1162 domain” to “VAB domain” throughout the revised manuscript according to the comment.

6. The authors found that two independent regions of Tsg101 (ie. UEV domain or PR-CC-SB domains) bound the DUF1162 domain. References were cited to suggest this bipartite binding has been observed previously, but in both examples given, a single domain of tsg101 (the UEV domain) bound two different regions of Hrs or Tal. This should be clarified in the text.

Thank you for underlying this deficiency. This section was revised, we have added the following explanation to clarified the bipartite interactions between VPS13D-TSG101: ‘In contrast to the bimodal interaction between VAB domain of VPS13D and UEV or PR-CC-SB domains of TSG101, the UEV domain of TSG101 independently interacted with the PR domain and C-terminal clathrin-binding domain of HGS (45).’

7. The authors state:” For instance, yeast VPS13 contributed to the growth of the prospore membrane (58) (59) by associating with spindle pole protein Cdc31 (60) “. This is not true: no direct link between the VPS13-Cdc31 interaction and prospore membrane growth has been demonstrated.

We thank the reviewer for catching the mistake. We rephrased the sentence according to the comment: ‘For instance, yeast VPS13 contributed to the growth of the prospore membrane, a double membrane which encapsulated the daughter nuclei to give rise to spores (58) (59).’

8. The discussion fails to discuss the current findings in the context of the relevant literature. Specifically, I would expect to see much more discussion of Anding et al 2018 (Current Biology 2018. 28:287). This paper identified VPS13D, Tsg101 and the ESCRTII component VPS36 in a genetic screen for mitophagy in drosophila, identified VPS13D as a ubiquitin-binding protein (via the UBA domain) and showed that reducing mitochondrial fusion rescued defects caused by loss of VPS13D. These results are similar to the findings presented in this paper yet have many key differences.

We appreciate your advice, and we rewritten the discussion section of this manuscript. We highlighted the elegant work by Anding et al. (Current Biology 2018. 28:287) by stating that the work by Anding et al. identified VPS13D as an ubiquitin binding protein and a key regulator of mitochondrial size in drosophila. We specifically discussed the potential connections between our work and the drosophila VPS13D study by Anding et al. including 1) ubiquitin binding of VPS13D; 2) loss of function effect of VPS13D on mitochondrial size.

9. In general, I found the figures hard to follow because so much of the data is presented in supplementary figures, the text does not always refer to the correct figure, panels are not always presented in order, and figure legends give few details (eg. don't always explain the markers used, don't always have a heading stating the key observation).

Thanks for this important comment. We have had the manuscript reorganized and the revised manuscript has 10 figures in the main text and 14 figures in supplementary materials. We corrected the typos in which the text failed to refer to the correct figure and swapped the original Fig.S12 and Fig.S13 to make them in correct order, the figure legends were rewritten.

Specific errors in figures include:

-Figure 2I, the second set of panels has “halo-VPS23” as a label. Is this a typo? Tsg101 is generally considered to be “mammalian VPS23”

Thanks for pointing out our mistake. We changed ‘halo-VPS23’ to ‘Halo-TSG101’.

-Figure S11H: in label for y-axis, what are GFP-EMCs? Any new abbreviations should be described in legend.

Thanks for the comment. The GFP-EMCs referred to fluorescence intensity of the reconstituted GFP in the ER-mitochondria MCS reporter cell line (SPLICS_{ER-mito} line). We now modified the y-axis labels to “Fluorescent intensity of reconstituted GFP (A.U.)”.

-Figures S12 and S13 appear to be swapped.

We have modified the order of the two figures according to your comment.

-The text says “our GFP-Trap assay using HEK293cells...identified...VAPB-PTPIP51 as potential VPS13D interactors (Fig. S7D)” yet S7D does not show GFP-Trap data.

We apologize for this error. We corrected the sentence to “Our GFP-Trap assay using HEK293 cells transfected with VPS13D^{ΔsfGFP} identified ER–mitochondrial MCSs tether complex VAPB-PTPIP51(56) as potential VPS13D interactors (Fig. S6B).

-In Fig4I, the spacing of the labels on the figure itself makes it hard to see which are mitochondrial vs ER

Thanks for the comment. We enlarged the labels and make it clearer.

-in Fig4H, the inset scale is too small to read.

Thanks for the comment. The scales on the insets of this figure were enlarged.

- In FigS8, I can't distinguish red vs pink in merged images. Pick different colors, or show each color before merge? A kymograph might be a better way to look at this data

Thank you for your kind advice. We are sorry for the inappropriate color combination in merged images. In revised manuscript, we modified the figure in two ways. First, we changed the pink 'LDs' to gray. We also did this in the corresponding movie (supplementary video-1). Second, we add a new merge panel in which Halo-TSG101 (labeled in green) and VAB-GFP (labeled in red) are shown. We also tried kymograph but the plot was way too noisy due to the 4 channels in the image.

- several figures (eg. 2F, S10D) use a t-test instead of ANOVA with post-hoc test, which would be more appropriate.

Thanks for your comment. In the revised manuscript, we conducted statistical analysis on the corresponding figures by one-way ANOVA with Turkey's multiple comparisons test.

Reviewer #2 (Remarks to the Author):

This study shows that over expressed VPS13D localizes to contact sites between lipid droplets. It identifies domains in VPS13D required for it to interact with both organelles. It identifies a domain VPS13D that interacts with the ESCRT component TGS101, which is required for VPS13D localization. It also presents evidence that contacts between lipid droplets and mitochondria are reduced when VPS13D is knocked down and that VPS13D is necessary for fatty acid transport to mitochondria. While these results are interesting, evidence that VPS13D plays a significant role in maintaining lipid droplet-mitochondria contacts and fatty acid transport is not well supported. There are number of concerns.

1. Lipid transport is assessed using BODIPY-labeled fatty acids. There are important concerns about whether the transport and metabolism of these fatty acids is similar to that of endogenous, unlabeled fatty acids. It is necessary to show that the movement and metabolism BODIPY-labeled fatty acids is similar to that of endogenous fatty acids. In addition, while the abstract suggests that the movement of fatty acids from lipid droplets to mitochondria is required for fatty acid oxidation, rates of oxidation have not been assessed. It has also not been shown that the BODIPY fatty acids are oxidized or even conjugated to CoA or carnitine, which is required for the uptake and oxidation of endogenous fatty acids by mitochondria. The rate of fatty acid transport should be directly assessed, and the results validated with endogenous fatty acids. Finally, there is no in vitro evidence that VPS13D can directly bind and transport fatty acids.

Response: Thanks for the comments. This comment can be further divided into two sub-questions, which were addressed separately.

1) Lipid transport is assessed using BODIPY-labeled fatty acids. There are important concerns about whether the transport and metabolism of these fatty acids is similar to that of endogenous, unlabeled fatty acids. It is necessary to show that the movement and metabolism BODIPY-labeled fatty acids is similar to that of endogenous fatty acids. In addition, while the abstract suggests that the movement of fatty acids from lipid droplets to mitochondria is required for fatty acid oxidation, rates of oxidation have not been assessed. It has also not been shown that the BODIPY fatty acids are oxidized or even conjugated to CoA or carnitine, which is required for the uptake and oxidation of endogenous fatty acids by mitochondria. The rate of fatty acid transport should be directly assessed, and the results validated with endogenous fatty acids.

We really appreciate that the reviewer pointed out the concerns about using fluorescent dye conjugated FAs. We agree that whether BODIPY-FA is a faithful marker for imaging of cellular FA trafficking is a valid question, but we are really sorry that we do not have the expertise to prove the faithfulness of BODIPY labeled FAs on basis of cellular FAs trafficking or oxidation.

However, we believe that the BODIPY-FAs are suitable imaging tools for tracing the movement of FAs in or between cells. Fluorescent-labeled FAs analogs were widely used in studying of FAs trafficking in the field, and have been shown to incorporate into LD-specific neutral lipids. These studies were listed below and incorporated into the reference list of the revised manuscript.

Thumser and Storch (2007) Mol. Cell. Biochem.

Wang and Lehner (2010) Mol. Biol. Cell

Herms and Pol (2013) Current Biology

Kassan and Pol (2013) Journal of Cell Biology

Rambold and Lippincott-Schwartz (2015) Developmental Cell

Chang and Lippincott-Schwartz (2019) Journal of Cell Biology

Ioannou, Lippincott-Schwartz and Liu (2019) Cell

The BODIPY 558/568-C₁₂ (Red C12) based pulse-chase assay was firstly developed in an elegant study conducted by Rambold and Lippincott-Schwartz (2015) Developmental Cell, and proved that Red C12 was incorporated into LDs in complete medium (no starvation) and was transferred to mitochondria (starvation) by imaging and the esterification of Red-C12 was further confirmed by thin layer chromatography (TLC).

In summary, we believe that BODIPY-C12 acts as a suitable tool for visualizing FAs trafficking. Regarding the concerns about oxidation of BODIPY conjugated FAs, it is again a valid question but we currently do not have the expertise to prove it. We believe that the oxidation of FAs required detailed study and is beyond the scope of this manuscript.

2) Finally, there is no *in vitro* evidence that VPS13D can directly bind and transport fatty acids.

This is a great question, and this question helps to improve the manuscript on the roles of VPS13D in FAs trafficking at mitochondria-LD MCSs. To further identify the potential role of VPS13D in FA transport, we carried out a series of study. VPS13D harbored a lipid transfer domain (LTD) at its N terminus, which was conserved to yeast VPS13 (De et al. (2017) JCB) and ATG2 (Valverde et al. (2018) JCB). As you have mentioned, two recent studies demonstrated that fungi VPS13 lipid transfer domain bound and transferred glycerophospholipids (Kumar et al. (2018) JCB; Li et al. (2020) JCB). To test whether VPS13D-LTD could directly bind lipid fatty acids

moieties *in vitro*, we performed protein-lipids co-migration assays. We found that, besides phospholipids, purified VPS13D-LTD could bind both medium-chain fatty acids C12 and long-chain fatty acid C16 (Fig.7A&B). Our work supported that VPS13D-LTP could directly bind FAs *in vitro*.

We further validated the role of the lipid transfer activities of VPS13D in FA transport in cells. Inspired by a recent study by Valverde et al (2019), in which a fragment of lipid transfer domain of ATG2A (mini-ATG2A, 1-345aa) can near-complete rescue the phenotype of ATG2 knockout upon overexpression (at level around 10 times endogenous), we tested whether VPS13D-LTD was sufficient for FA transfer from LD to mitochondria by rescuing the FA transfer defects induced by VPS13D suppression. Based on the high conservation between VPS13 and ATG2 at their N terminal lipid transfer domain, we made a hydrophobic surface LTD mutant (VPS13D-LTD mutant: L42R , L66R , L85R , I148R , I152E , I174R , M181R , F203E , L278R, L293R , W319R) (Fig. 7C), which could still bind but could not transport lipid fatty acids moieties (Valverde et al. (2018) JCB; Li et al. (2020) JCB). As shown in Fig.7G, VPS13D-LTD substantially rescued the VPS13D suppression induced FAs trafficking defects upon 5 h or 24 h EBSS starvation. Instead the VPS13D-LTD mutant did not rescue the FAs trafficking defects (Fig.7H), as indicated by colocalization between red C12 and mitochondrial marker via the Pearson's correlation coefficient (Fig.7C). We also found that the rescue effects of VPS13D-LTP were substantially less than that of full-length protein, though the VPS13D-LTD was highly expressed by a CMV promoter. This result suggested that, besides the lipid transfer activities, the VPS13D-TSG101 interactions also played an important role in this process. In summary, our results supported an important role of VPS13D-LTD in FAs trafficking at mitochondria-LD MCSs.

The previous study by Kumar (2018) JCB showed that yeast Vps13 α bound to glycerophospholipids but not other lipid species in Expi 293F cells by liquid chromatography–tandem mass spectrometry (LC/MS/MS). We postulated that there were two reasons to explain why FAs were not identified with co-purified VPS13 α by LC/MS/MS in the previous work. First, the ionization efficiency of FAs under electrospray ionization mass spectrometry conditions is poor, rendering the construct of a robust platform for profiling of various FAs in biological samples using LC-MS-MS very challenging (Jun Mok et al (2016) RSC Advances). Besides, the identification of FFAs is also easily affected by matrix effects and the sensitivity is low (Narayana et al. (2015) Chemistry & Biology). Second, the release of FAs from triglyceride by lipases only occurred upon stimulation, for instance, starvation or

exercise. Consistently, our results showed that red C12 was barely mobilized and transferred from LDs to mitochondria in complete medium (Figs.7E-I). Therefore, we postulated that the low abundance of FAs in cells cultured in complete medium might be another reason

These results were shown below and incorporated into Fig. 7 in the revised manuscript.

Figure 7

2. The images in Fig. S9B suggest that there are mitochondria that are not in contact with lipid droplets. Is the rate of fatty acid transport to these mitochondria less than those in contact with lipid droplets?

This is again an insightful question. We analyzed the level of FAs transfer to mitochondria contacting LDs versus mitochondria not adjacent to LDs (2 μm away from any LDs for 2min imaging) upon 5h starvation by confocal microscopy (Fig.S9A). We found the FAs level in mitochondria contacting LDs was substantially higher than that of mitochondria not adjacent to LDs, as revealed by the fluorescent intensity of Red C12 in mitochondria (Fig.S9B). We also observed that the Red C12 appeared to be more efficiently transferred to mitochondria in peri-nuclear regions, where LDs formed extensive MCSs with mitochondria.

These results were shown below and incorporated into Fig. S9 in the revised manuscript.

Figure S9

3. The localization of VPS13D at lipid droplet-mitochondria junctions is only seen with overexpressed protein. The paper would be stronger if could be shown that endogenous VPS13D is also at these contact sites. I realize this is a challenging problem because VPS13D is a low abundance protein.

Thank you for raising this important point, and we really appreciate the understanding of the technical difficulties in making Crispr-cas9 mediated knock-in cells for low abundance/large molecular weight protein such as VPS13D. During revision, we made an extensive effort on this KI cell lines. Although we successfully inserted GFP into the site of VPS13D gene, same as our VPS13D^{sfGFP}, we just could not observe any generic fluorescent signals (data not shown).

Alternatively, we tried to use a commercial VPS13D polyclonal antibody to stain the endogenous VPS13D in HEK293 cells by immunofluorescence staining (IF). Although this antibody was not initially effective in IF, its specificity for VPS13D could be improved by preclearing the antibody against fixed and permeabilized cells from VPS13D depleted cells according to the protocol modified from an elegant work (Valverde et al. (2018) JCB) (Fig.S1C). This pre-cleared antibody was effectively localized to mitochondria-LDs junctions under OA/EBSS but its localization at these junctions was near-complete lost in VPS13D depleted cells (Fig.S1D).

These results were shown below and incorporated into Fig. S1 in the revised manuscript.

Figure S1

4. The evidence that knockdown of VPS13D reduces lipid droplet-mitochondria contacts is not strong. Using split-GFP is not a good method to assess contacts since, as the authors point out, the interaction is irreversible. No controls are included to show that the proteins are expressed at the same level in the different cell lines. The images in Fig. S9 suggest that most mitochondria remain in contact with lipid droplets after VPS13D knockdown. There are fewer contacts per mitochondria length but also, it seems, fewer lipid droplets. There is no evidence that fewer contacts reduces lipid transport rates.

Response: Thanks for the comments. This comment can be further divided into three sub-questions, which were addressed separately.

1) The evidence that knockdown of VPS13D reduces lipid droplet-mitochondria contacts is not strong. Using split-GFP is not a good method to assess contacts since, as the authors point out, the interaction is irreversible. No controls are included to show that the proteins are expressed at the same level in the different cell lines.

Thank you for raising this important question. We agree that split-GFP is not optimal for MCSs assessment. Therefore, we modified the split-GFP method in two ways to make it more suitable for our purpose. First, the expression of splitGFP was under control of Tet-on promoter. Second, the two splitGFP components GFP₁₋₁₀ and GFP11 were expressed in one mRNA transcript, thus making the ratio of GFP₁₋₁₀ to GFP11 as 1:1. We carefully titrated the induction conditions of splitGFP to ensure the normal morphologies of mitochondria and LDs and no dramatic artificial tethering.

During revision, we examined the level of GFP₁₋₁₀ in both scrambled or VPS13D siRNA treated cells by immunoblots analysis, and found that the GFP₁₋₁₀ level was not substantial changed upon VPS13D depletion (Fig. S8&E).

These results were shown below and incorporated into Fig. S8 in the revised manuscript.

Figure S8

In addition, we further confirmed the role of VPS13D in mitochondria-LD junctions by using reversible dimerization-dependent fluorescent proteins (ddFPs) system, consisting of a heterodimeric fluorescent protein complex made up of a dim GFP (GA) and protein-binding partner (B) that lacks a chromophore. Upon dimerization of with B, the brightness of GA was enhanced. GA was fused to Tom20 to target it to OMM, and the B part was fused to LD membrane protein Plin2 to target it to the LD. We observed that the GFP signals were specifically enriched at mitochondria-LDs junctions though we indeed observed that its fluorescent signals were weaker compared to the splitGFP signals, possibly due to the reversible feature of GA-B

dimerization (Fig. S8I). Suppression of VPS13D significantly reduced the mitochondria-LDs interactions but TSG101 depletion did not substantially alter such MCSs, as revealed by flow cytometry of GFP fluorescence (Fig. S8J).

These results were shown below and incorporated into Fig. S8 in the revised manuscript.

Figure S8

2) The images in Fig. S9 suggest that most mitochondria remain in contact with lipid droplets after VPS13D knockdown. There are fewer contacts per mitochondria length but also, it seems, fewer lipid droplets. There is no evidence that fewer contacts reduces lipid transport rates.

Thanks for this comment.

Our results clearly showed that there were mitochondria contacting LDs in VPS13D depleted cells, but the extent of these interactions were substantially less (Fig. 5E&F). In addition, by time-lapse imaging, we found that the mitochondria-LD contacts were more transient in VPS13D depleted cells (Fig.5J&K).

The reduction in mitochondria-LD interactions by VPS13D depletion was not due to fewer lipid droplets in the cells because our results indicated that VPS13D suppression resulted in more LDs (number of LDs /cell area) (Fig.5I).

In summary, our results support that VPS13D was required for maintenance of mitochondria-LD interactions.

The original Fig. S9B was moved to Fig. 5E.

5. While the results in Figure 4 are interesting, it is not clear that they add to the central point of this study.

Thanks for the comment. First, mitochondria, LD and endoplasmic reticulum (ER) were physically associated and formed a tri-organelle interactions (Freyre et al. (2019) Molecular Cell). It is still unknown how the tripartite MCSs are coordinated. Our work demonstrated a novel function of VPS13D in FAs trafficking at mitochondria-LD MCSs, and its role in negative regulation of the other MCSs, ER-mitochondria MCSs. Since cross talk between different MCSs has been a hot topic in the field, our results in Figure 4 would shed lights on the coordination of the tripartite MCSs, and enhanced the impact of our manuscript.

On the other hand, ER played a prominent role in lipids metabolism. Since our results showed VPS13D was directly involved in FAs trafficking, we then moved on to explore whether VPS13D suppression affected ER. Indeed, we found a dramatic increase in interactions between ER and mitochondria. Currently the biological significance of connections between these two MCSs was unclear. We hypothesized that the elevation of ER-mitochondria interactions may compensate the loss of mitochondria-LD connections in FAs trafficking, which needs to be studied in detail in future.

6. It is not clear whether the authors are proposing that the ESCRT proteins are part of a tether or whether play a direct role in fatty acid transport. The second is a more interesting idea, but there is no evidence to support it.

We thank the reviewer for the insightful comment. We have in fact carried out a study on this question and found that the suppression of ESCRT protein TSG101 did not substantially alter the mitochondria-LD interactions in ddGFP-based reporter cells (Fig. S8I&J). Therefore we agree with the reviewer that TSG101 and possibly other ESCRTs proteins do not act as a tether.

Our results support essential and direct roles of ESCRT proteins in remodeling of LD morphologies in cooperation with VPS13D.

First, our results showed that TSG101 was recruited to LDs upon OA/EBSS stimulation in HEK293 cells (Fig. 4G). We observed that LDs underwent morphological changes (budding/tubulation in the inset in Fig. 4G) in some LDs.

Second, co-expression the VAB domain of VPS13D and TSG101 strongly remodeled the LD morphologies, either by constricting LDs (Fig. 4I, insets 1, 3, 4, 6) or by budding/tubulation from LDs (Fig. 4I, insets 2, 5, 6) with specific co-localization of VAB-GFP and Halo-TSG101 at constrictive sites, as shown by high-resolution Leica lightning live cell microscopy, suggesting a direct role of VAB domain of VPS13D and TSG101 in modulating LD morphology.

Third, suppression of ESCRT proteins including TSG101, CHMP4B, CHMP6, CHMP1B and IST1 significantly inhibited the Red C12 transfer rate from mitochondria to LDs upon starvation, indicating ESCRTs were required in this process.

Fourth, the mechanistic relationships between remodeling of LD membranes and FAs trafficking are unclear so far (Vietri and Stenmark (2019) Nature Review). We hypothesized that remodeling of LD membranes facilitated the access of tri-glyceride (inside LDs) to lipase (cytosol or cytoplasmic face of LD membrane), thus enhancing the FA trafficking from LDs to mitochondria. Detailed work needs to be done to test this hypothesis in future.

In summary, our results supported the direct role of ESCRTs in remodeling of LD membrane morphologies. Our results were consistent with a previous study by Chang et al. (2019).

Fig. 4G & I (originally Fig. 2 G & I) were shown below.

Reviewer #3 (Remarks to the Author):

This is a paper that presents interesting results that will be of sufficient interest and novelty for consideration in Nature Comms. A potential new function for ESCRT complex components. The problem for me is the conveyance of the message - it is a sprawling, disengaging read. For my mind there is way too much supplementary material- (16 figures) vs 4 in the main text. Even the summary model is a supplementary figure. I think the authors should consider a major overhaul of the manuscript to make it more direct and succinct- concentrating on the firmest data. This does not include their ubiquitin binding data which are particularly opaque and rely on constructs that have had multiple lysines mutated, which is not clearly explained. The paper is largely descriptive and one is left wondering how the observed phenomena relate to the near universal property of ESCRT complexes in constriction and severance of membranes. Error bars are not always clearly explained and some data handling is a bit unclear. For example data presented in Fig 1C and D , cell counts are given, but does this amalgate Biological repeats?

We are deeply grateful for your support of our work. Your insightful suggestions help us re-organize the manuscript and highlight the impact of our work. The comment can be further divided into four sub-questions, which were addressed separately.

1) The problem for me is the conveyance of the message - it is a sprawling, disengaging read. For my mind there is way too much supplementary material- (16 figures) vs 4 in the main text. Even the summary model is a supplementary figure. I think the authors should consider a major overhaul of the manuscript to make it more direct and succinct- concentrating on the firmest data.

We are sorry for the structure and figure issues in our manuscript. According to your and other reviewers' comments, we reorganized the manuscript and make it clearer. In the revised manuscript there are 10 figures in the main text (nine data figures and one figure of working model) and 14 figures in the supplementary material.

2) This does not include their ubiquitin binding data which are particularly opaque and rely on constructs that have had multiple lysines mutated, which is not clearly explained.

We are sorry for the opaqueness in description of the two ubiquitin mutants in our manuscript. HA-K63-linked ubiquitin mutant only harbored lysine 63 and other lysines were mutated to arginines, and HA-K48-linked ubiquitin mutant only contained lysine 48 with other lysines were mutated to arginines.

3) The paper is largely descriptive and one is left wondering how the observed phenomena relate to the near universal property of ESCRT complexes in constriction and severance of membranes.

This is an insightful question. We discussed the potential mechanistic connections between FAs trafficking and the constricting and severing activities of ESCRT complex on LD membrane in the discussion section of revised manuscript.

ESCRTs were considered to function in inverse budding reactions, in which ESCRT-III proteins form filaments that assemble on membranes with a negative curvature (35). However, recent studies revealed that ESCRTs could also mediate classical-topology membrane remodeling, in which the ESCRTs formed a double-stranded helical polymer that coats positively curved membranes (36). For instance, IST1 promotes fission of endosomal tubules via recruitment of the microtubule-severing ATPase spastin to mediate endocytic recycling(38), and both IST1 and CHMP1B mediate fatty acid transfer from lipid droplets to peroxisomes, possibly by modifying the lipid droplet morphology (44).

Our results further supported a direct role of ESCRT complexes in FAs trafficking from LDs to mitochondria through remodeling LD morphologies. First, our findings showed that ESCRT-I protein TSG101 can be recruited to LDs under OA/EBSS (Fig. 4F,G). We noticed that TSG101 decorated LDs underwent morphological remodeling (membrane tubulation, Fig. 4G, inset). In cells with co-expression of VAB domain of VPS13D and TSG101 upon OA/EBSS stimulation, the morphology of LD membrane was strikingly remodeled, either by constricting LDs (Fig. 4I, insets 1, 3, 4, 6) or by budding/tubulation from LDs (Fig. 4I, insets 2, 5, 6) with co-localization of VAB-GFP and Halo-TSG101 at specific constrictive sites, suggesting a direct role of VAB domain of VPS13D and TSG101 in modulating LD morphology.

Due to the unique phospholipid monolayer of LDs and the spatial segregation between triglyceride (inside LDs) and lipase (associated with cytoplasmic face of LD membrane), it was plausible that such remodeling of LD morphology by TSG101/other ESCRT components promoted the access of triglyceride to membrane-associated lipase, and eventually facilitated triglyceride mobilization by lipase.

Fig. 4G & I (originally Fig. 2 G & I) were shown below.

3) Error bars are not always clearly explained and some data handling is a bit unclear. For example data presented in Fig 1C and D, cell counts are given, but does this amalgate Biological repeats?

Thanks for the comment. All the error bars were re-plotted with clear explanations in the statistical analysis section of revised manuscript and specific statistic analysis was described in the legend in the revised manuscript.

All of the quantifications performed in this study were based on the pools of data from at least three independent experiments.

Quantification raw data of the revised manuscript (source data file. xlsx) were submitted along with this revised manuscript.

REVIEWER COMMENTS

Reviewer #1 (Remarks to the Author):

The authors have done an impressively thorough job of addressing my questions and concerns. Importantly, they have added a number of new experiments that have enhanced the manuscript, and done extensive re-writing. I now feel this work is acceptable for publication.

Reviewer #2 (Remarks to the Author):

While some of my concerns have been addressed, there are still a number of important issues.

1. Additional evidence is still required to support the claim that VPS13D traffics fatty acids, which is one of the central conclusions. My previous review raised concerns about using BODIPY-fatty acids to measure trafficking of endogenous fatty acids, particularly whether they are good proxies for endogenous fatty acids. The authors responded that others have used BODIPY-labeled lipids to measure lipid trafficking and that they do have the expertise to determine whether BODIPY-labeled lipids are good proxies for endogenous lipids. Both points are correct but irrelevant. The text should have an acknowledgement and discussion of the well-known caveats about using lipids labeled with fluorescent groups like BODIPY to estimate the movement of unlabeled lipids. In addition, for this study to be appropriate for Nature Communications or a similar journal, it is necessary to confirm some of the findings by measuring the metabolism of unlabeled fatty acids. There are well established methods to directly determine the rates of fatty acid incorporation to mitochondrial membranes and fatty acid oxidation in mitochondria. Performing these assays will help confirm the finding with BODIPY fatty acids.
2. To confirm the claim that VPS13D binds fatty acids, the stoichiometry of binding should be estimated, and it should be possible to compete the bound BODIPY fatty acid with unlabeled fatty acids.
3. The results in Fig. 6JK should be normalized to the total amount of BODIPY fatty acid taken up.
4. Pearson's coefficient is not an appropriate way to estimate the rate of BODIPY fatty acid transport to mitochondria (Fig. 6L) since the coefficient is affected by factors unrelated to transport such as the total amount of BODIPY fatty acid taken up and the size of mitochondria. The absolute rate of increase in the BODIPY signal in mitochondria should be determined.
5. Fig S1 should be a main figure and localization of overexpressed VPS13D (Fig. 1) should be moved to supplemental data or removed. It would be nice if the percent of LD-mitochondria junctions containing endogenous VPS13D were determined. In addition, the quantification in Fig. 1D cannot be correct; the inset in Fig. 1F shows most puncta are far from mitochondria (it is also not clear what counts as one puncta since they are sometimes connected).
6. The claim that LDs with VAB/TSG101-decorated LDs are constricted or are budding or tubulating, which would be an important new finding, is not well supported by the fluorescence microscopy shown (Fig. 4I). The possibility that ESCRT proteins deform LDs needs to be verified by EM.
7. The manuscript is very dense and hard to understand. As Reviewer 3 suggested, the authors should consider focusing on one main story and removing unnecessary data. In addition, the logic of the experiments is often not well explained. There are some mistakes in the figures, for example the times in Fig. S2A are not shown and some figure legends do not explain what the arrows indicate. The manuscript needs to be carefully edited.

Reviewer #3 (Remarks to the Author):

The manuscript remains a long and challenging read but contains much data that is of interest.

Point-by-point response to Reviewer's Comments

Reviewer #1 (Remarks to the Author):

The authors have done an impressively thorough job of addressing my questions and concerns. Importantly, they have added a number of new experiments that have enhanced the manuscript, and done extensive re-writing. I now feel this work is acceptable for publication.

Response: We are truly grateful to you for your constructive comments and insightful suggestions, which significantly improve the quality of this manuscript.

Reviewer #2 (Remarks to the Author):

While some of my concerns have been addressed, there are still a number of important issues.

1. Additional evidence is still required to support the claim that VPS13D traffics fatty acids, which is one of the central conclusions. My previous review raised concerns about using BODIPY-fatty acids to measure trafficking of endogenous fatty acids, particularly whether they are they are good proxies for endogenous fatty acids. The authors responded that others have used BODIPY-labeled lipids to measure lipid trafficking and that they do have the expertise to determine whether BODIPY-labeled lipids are good proxies for endogenous lipids. Both points are correct but irrelevant. The text should have an acknowledgement and discussion of the well-known caveats about using lipids labeled with fluorescent groups like BODIPY to estimate the movement of unlabeled lipids. In addition, for this study to be appropriate for Nature Communications or a similar journal, it is necessary to confirm some of the findings by measuring the metabolism of unlabeled fatty acids. There are well established methods to directly determine the rates of fatty acid incorporation to mitochondrial membranes and fatty acid oxidation in mitochondria. Performing these assays will help confirm the finding with BODIPY fatty acids.

Response: Thanks for the comments. These comments can be further divided into two sub-questions, which were addressed separately.

1)The text should have an acknowledgement and discussion of the well-known caveats about using lipids labeled with fluorescent groups like BODIPY to estimate the movement of unlabeled lipids.

Response: Thanks for the comment. We have acknowledged the concerns of fluorophore-conjugated lipids in lipid trafficking in the result section of revised manuscript. These sentences were shown below.

However, it should be noted that the fluorophore-conjugated FAs likely have different biophysical properties from the endogenous FAs, which could affect on their trafficking and metabolism (Maekawa et al. (2014) Journal of Cell Science), the *in vivo* FA trafficking

results using BODIPY-conjugated FAs (Red C12) should be further verified with the use of isotopically labeled FAs.

2) In addition, for this study to be appropriate for Nature Communications or a similar journal, it is necessary to confirm some of the findings by measuring the metabolism of unlabeled fatty acids. There are well established methods to directly determine the rates of fatty acid incorporation to mitochondrial membranes and fatty acid oxidation in mitochondria. Performing these assays will help confirm the finding with BODIPY fatty acids.

Response: Thanks for the insightful suggestion.

As an alternative approach, we confirmed the roles of VPS13D and TGS101 in the transport and oxidation of endogenous FAs in mitochondria via Seahorse FA oxidation (FAO) assays. siRNA-mediated suppression of either VPS13D or TGS101 in HepG2 cells cultured in substrate-limited medium led to a slight reduction in basal respiration but a higher reduction in the maximal respiration of mitochondria. As a control, cells were pre-treated with Etomoxir (Eto, a carnitine palmitoyl transferase-1 inhibitor), which substantially reduced both the basal and maximal respiration of mitochondria (Fig. 6M). The reduced FA oxidation observed in VPS13D-suppressed cells was consistent with the findings of two clinical studies, in which cells from patients of ataxia with spasticity (caused by loss-of-function mutations of VPS13D) showed lipodosis and reduced energy productions (Gauthier. et al. (2018) *Annals of Neurology*) (Seong. et al. (2018) *Annals of Neurology*). In addition, a previous study by Anding. et al. showed that the cristae of mitochondria appeared to be intact in VPS13D-depleted cells, as observed by TEM, indicating that the capacity of mitochondria for energy generation might be largely unaffected upon VPS13D-suppression (Anding. et al (2018) *Current Biology*). Therefore, these results suggest that the low energy production and lipodosis observed in the patient cells may be attributed to the impaired transfer of FAs to mitochondria.

This result was shown below and incorporated into the revised manuscript as Fig. 6M.

M
2. To confirm the claim that VPS13D binds fatty acids, the stoichiometry of binding should be estimated, and it should be possible to compete the bound BODIPY fatty acid with unlabeled fatty acids.

Response: Thanks for this insightful suggestion.

As suggested, to confirm that VPS13D binds FAs, we sought to estimate the stoichiometry of the binding of VPS13D-LTD to FAs via in vitro protein-lipids binding assays. Through titration, we found that one VPS13D-LTD molecule (320 residues) could bind ~5 molecules of BODIPY-C16 (Fig. 7C, D), and we further estimated that a full-length VPS13D molecule with a long groove serving as a lipid transfer channel at its N-terminus (~1500 residues) could bind ~25 molecules of BODIPY-C16, which agreed with the estimated stoichiometry of the binding of yeast VPS13 and ATG2 to phospholipids (17, 27), and this result supported the notion that VPS13 is a bridge lipid transporter that is capable of binding tens of lipids at once (53). In addition, to validate the finding that VPS13D binds endogenous, unlabeled FAs, we examined whether unlabeled FAs could outcompete BODIPY-C16 for binding to VPS13D-LTD. Competition assays indicated that palmitic acid, but not OA, could sufficiently outcompete BODIPY-C16 for binding to VPS13D-LTD in a dose-dependent manner (Fig. 7E, F). Collectively, these results supported the hypothesis that VPS13D-LTD directly binds FAs in vitro.

These results were shown below and incorporated into the revised manuscript as Fig. 7C, D and Fig. 7E, F, respectively.

3. The results in Fig. 6JK should be normalized to the total amount of BODIPY fatty acid taken up.

Response: Thanks for the comment. We normalized the Red C12-labeled LD size and number to total fluorescence of BODIPY-C12 in each cell. The new quantification results were shown below, and incorporated in the present manuscript as Fig. 6J and Fig. 6K, respectively.

4. Pearson's coefficient is not an appropriate way to estimate the rate of BODIPY fatty acid transport to mitochondria (Fig. 6L) since the coefficient is affected by factors unrelated to transport such as the total amount of BODIPY fatty acid taken up and the size of mitochondria. The absolute rate of increase in the BODIPY signal in mitochondria should be determined.

Response: Thanks for the comment. We quantified the level of BODIPY-C12 uptake to mitochondria by measuring the total fluorescence intensity of BODIPY-C12 inside mitochondria. The method for this quantification was described in the methods and materials section in the present manuscript.

The new quantification results were shown below and incorporated in the present manuscript as Fig. 6L and Fig. 7G, respectively.

5. Fig S1 should be a main figure and localization of overexpressed VPS13D (Fig. 1) should be moved to supplemental data or removed. It would be nice if the percent of LD-mitochondria junctions containing endogenous VPS13D were determined. In addition, the quantification in Fig. 1D cannot be correct; the inset in Fig. 1F shows most puncta are far from mitochondria (it is also not clear what counts as one puncta since they are sometimes connected).

Response: Thanks for the comments. This comment can be further divided into three sub-questions, which were addressed separately.

1) Fig S1 should be a main figure and localization of overexpressed VPS13D (Fig. 1) should be moved to supplemental data or removed.

Response: We moved the original Fig.S1C to main figures in Fig. 1 and move the original Fig.1 to supplemental data (Fig. S1). In addition, we deleted the original figures including Fig. 1A, B, C & D to make this set of results clearer.

2) It would be nice if the percent of LD-mitochondria junctions containing endogenous VPS13D were determined.

Response: We thank reviewer for this helpful suggestion. We made two quantifications of the distribution of the endogenous VPS13D puncta from IF images. First, we measured the percentage of VPS13D relative to mitochondria-LD junctions. We found that around half (~49%) of VPS13D puncta (n=644) localized to mitochondria-LD junctions (Fig. 1E). Second, we examined the percentage of mitochondria-LD junctions with the presence of VPS13D puncta, and we found ~79% of well-resolved mitochondria-LD junctions (n=286) with VPS13D puncta (Fig. 1F).

These results were shown below and incorporated in the present manuscript as Fig. 1E and Fig. 1F, respectively.

E

F

3) In addition, the quantification in Fig. 1D cannot be correct; the inset in Fig. 1F shows most puncta are far from mitochondria (it is also not clear what counts as one puncta since they are sometimes connected).

Response: We appreciate the reviewer to point out this confusion in the quantifications of VPS13D^{ΔsfGFP} puncta. We had only counted the readily resolved, discrete VPS13D^{ΔsfGFP} puncta in this quantification, and we did not take into account the VPS13D 'sheet' (lots of puncta interconnected to a 'sheet'), which may be attributed to artifacts of overexpression. We have removed this set of quantification results (the original Fig. 1B and C) to avoid confusion.

6. The claim that LDs with VAB/TSG101-decorated LDs are constricted or are budding or tubulating, which would be an important new finding, is not well supported by the

fluorescence microscopy shown (Fig. 4I). The possibility that ESCRT proteins deform LDs needs to be verified by EM.

Response: We thank the reviewer for appreciating this piece of data and also for this insightful suggestion, which substantially enhance the impact of our manuscript.

To further confirm the roles of the VAB domain of VPS13D and TSG101 in the remodeling of LD membranes, we directly examined the LD morphologies upon OA/EBSS stimulations by performing transmission electron microscopy (TEM) of HEK293 cells co-expressing VAB-GFP and Halo-TSG101. Electron micrographs revealed that the shape of the LDs was dramatically altered upon overexpression of Halo-TSG101 and VAB-GFP: LDs underwent either budding (Fig. 4I, right panel; inset 1) or constriction (Fig. 4I, right panel; inset 2). However, the LDs in OA/EBSS-treated cells expressing the empty vectors were typically spherical with mild deformations (Fig. 4I, left panel). In addition, we noted that the mitochondria were highly condensed in the VAB/TSG101-overexpressing cells, suggesting that the mitochondrial functions might be impaired in these cells.

These results were shown below and incorporated into the revised manuscript as Fig. 4I.

7. The manuscript is very dense and hard to understand. As Reviewer 3 suggested, the authors should consider focusing on one main story and removing unnecessary data. In addition, the logic of the experiments is often not well explained. There are some mistakes in the figures, for example the times in Fig. S2A are not shown and some figure legends do not explain what the arrows indicate. The manuscript needs to be carefully edited.

Response: Thanks for the comments. This comment can be further divided into two sub-questions, which were addressed separately.

1)As Reviewer 3 suggested, the authors should consider focusing on one main story and removing unnecessary data.

Response: We thank reviewers for this helpful suggestion. As suggested, we focused on one main story, the role of ESCRTs and VPS13D in the remodeling of LD membranes and subsequent transfer of FAs. In the present manuscript, we deleted the results about the regulation of ER-mitochondrial interactions by VPS13D and other unnecessary data.

Specifically, the following figures and their corresponding texts were removed in the present manuscript.

1. Some unnecessary data of the localizations of VPS13D^{ΔsfGFP}. (Original Fig. 1A, B, C & D; Fig. S1A)
2. Some unnecessary data of the identifications of two amphipathic helix for LD targeting. (Original Fig. S2C, D & E)
3. The data of the identification of FFAT motif in VPS13D (Original Fig. S3).
4. The unnecessary data of the interactions between K63-linked ubiquitin and VAB domain (Original Fig. S4)
5. The data of the regulation of ER-mitochondrial interactions by VPS13D (Original Fig. 8, Fig. 9 and Figs. S10, S11, S12, S13, S14 and S15).

In addition, to make this manuscript easier to read, we have changed the positions of two figures. Specifically, the original Fig. S7I (the dynamics of TSG101 and VAB at mitochondrial-LD MCSs) have been moved to the main figures as Fig.7J in the present manuscript. Additionally, the original Fig.7C (sequence alignment of LTD between VPS13D and ATG2) has been moved to the supplementary figures as Fig. S7C in the present manuscript.

2) In addition, the logic of the experiments is often not well explained. There are some mistakes in the figures, for example the times in Fig. S2A are not shown and some figure legends do not explain what the arrows indicate. The manuscript needs to be carefully edited.

Response: We thank reviewer for catching the missing of the timestamps on Fig.S2A and the missing of explanations of arrows in some figures. In the present manuscript, we added the timestamps in Fig.S2A and clearly explained all arrows in each figure. In addition, we carefully edited the manuscript and enhanced the logical flow of the manuscript.

Reviewer #3 (Remarks to the Author):

The manuscript remains a long and challenging read but contains much data that is of interest.

Response: We are sincerely grateful to you for both your interests on our findings and your great suggestions on the structure of this manuscript. To make this manuscript concise, we have focused on one main story, the role of ESCRTs and VPS13D in the remodeling of LD membranes and subsequent transfer of FAs, in the present manuscript. To do so, we have deleted the results about the VPS13D-mediated regulation of ER-mitochondrial interactions and other unnecessary data.

Specifically, the following figures and their corresponding texts were removed in the present manuscript.

1. Some unnecessary data of the localizations of VPS13D^{ΔsfGFP}. (Original Fig. 1A, B, C & D; Fig. S1A)
2. Some unnecessary data of the identifications of two-amphipathic helix for LD targeting. (Original Fig. S2C, D & E)
3. The data of the identification of FFAT motif in VPS13D (Original Fig. S3).
4. The unnecessary data of the interactions between K63-linked ubiquitin and VAB domain (Original Fig. S4)
5. The data of the regulation of ER-mitochondrial interactions by VPS13D (Original Fig. 8, Fig. 9 and Figs. S10, S11, S12, S13, S14 and S15).

In addition, to make this manuscript easier to read, we have changed the positions of two figures. Specifically, the original Fig. S7I (the dynamics of TSG101 and VAB at mitochondrial-LD MCSs) have been moved to the main figures as Fig.7J in the present manuscript. Additionally, the original Fig.7C (sequence alignment of LTD between VPS13D and ATG2) has been moved to the supplementary figures as Fig. S7C in the present manuscript.

REVIEWERS' COMMENTS

Reviewer #2 (Remarks to the Author):

My concerns have largely been addressed. There are a few minor issues.

1. The seahorse data are a nice addition, but they do not resolve the issue of whether Vps13D directly transports fatty acids to mitochondria for oxidation. Considered by themselves, the seahorse data says only that depletion of Vps13D modestly compromises fatty acid oxidation, but do not indicate whether Vps13D directly transports fatty acids. Without a demonstration that BODIPY fatty acids are metabolized like unlabeled fatty acid or a direct measurement of unlabeled fatty acid transport, I think it remains possible that Vps13D mediates LD-mito contacts but does not directly transport fatty acids. I think this possibility should be discussed.
2. The EM in Fig. 4i strengthen the claim that the ESCRT proteins deform LDs. It would be good if the images could be quantified, either by determining the percent of LDs that are not spherical or, even better, by determining how significantly LDs vary from being spherical in both the cells expressing VAB-GFP and Halo-TSG101 and the control cells. I do not think the images make a strong case that LDs bud or tubulate LDs, though they might.
3. The quantifications presented in Fig. 6j and 6k are confusing. What does it mean that LD size and number were "normalized to C12 fluorescence?" When I requested that the data in 6j and 6k be normalized to amount of fatty acid taken up, I wondered if there was a correlation between the size and number of LDs and the amount of fatty acid taken up. For example, there is an increase in the size and number of LDs in cells depleted of VPS13D. Is this because there is less fatty acid movement from LDs (as the study suggests) or because the VPS13D-depleted cells took up more fatty acid than the control cells? This should be clarified.
4. Fig. 6l should include a column for cells depleted of VPS13D.
5. The data in Fig. 6b-d should include number of pmoles (or nmoles) of protein and lipids instead of ratios of fluorescence.

Point-by-point response to Reviewer's Comments

Reviewer #2 (Remarks to the Author):

My concerns have largely been addressed. There are a few minor issues.

We sincerely appreciate all the insightful comments from reviewer, which substantially improved the manuscript.

1. The seahorse data are a nice addition, but they do not resolve the issue of whether Vps13D directly transports fatty acids to mitochondria for oxidation. Considered by themselves, the seahorse data says only that depletion of Vps13D modestly compromises fatty acid oxidation, but do not indicate whether Vps13D directly transports fatty acids. Without a demonstration that BODIPY fatty acids are metabolized like unlabeled fatty acid or a direct measurement of unlabeled fatty acid transport, I think it remains possible that Vps13D mediates LD-mito contacts but does not directly transport fatty acids. I think this possibility should be discussed.

We thank reviewer for this helpful comment. To clarify this point, we added the following sentences in the discussion section of the revised manuscript.

'In addition, it should be noted that the results from the seahorse assays only indicated that VPS13D depletion reduced the oxidation of FAs (Fig. 6m), but did not indicate a direct role of VPS13D in the transfer of FAs. Although our results demonstrated that the LTD of VPS13D bound FAs *in vitro* and was required for the efficient transfer of Red C12 to mitochondria *in vivo*, more direct evidence is required to confirm that VPS13D directly transports endogenous FAs at MCSs.'

2. The EM in Fig. 4i strengthen the claim that the ESCRT proteins deform LDs. It would be good if the images could be quantified, either by determining the percent of LDs that are not spherical or, even better, by determining how significantly LDs vary from being spherical in both the cells expressing VAB-GFP and Halo-TSG101 and the control cells. I do not think the images make a strong case that LDs bud or tubulate LDs, though they might.

Thanks for this insightful comment. As suggested, we added the quantifications for the EM images and the quantifications were incorporated in the revised manuscript as Fig. 4j, and were shown below. In the quantifications, we manually categorized the LDs into normal and deformed, based on their shapes. Briefly, normal LDs were those with a typical spherical shape as shown in the left panel of Fig.4i; the deformed LDs were those with evident morphological alterations as shown in the right panels of Fig. 4i.

3. The quantifications presented in Fig. 6j and 6k are confusing. What does it mean that LD size and number were “normalized to C12 fluorescence?” When I requested that the data in 6j and 6k be normalized to amount of fatty acid taken up, I wondered if there was a correlation between the size and number of LDs and the amount of fatty acid taken up. For example, there is an increase in the size and number of LDs in cells depleted of VPS13D. Is this because there is less fatty acid movement from LDs (as the study suggests) or because the VPS13D-depleted cells took up more fatty acid than the control cells? This should be clarified.

We thank reviewer for pointing out the confusion in the quantifications in Fig. 6j and 6k. To test whether cells with the depletion of VPS13D or ESCRT proteins took up more fatty acids than control cells, we estimated the total amounts of BODIPY-C12 at t=0 (right before starvation) in control or VPS13D, ESCRT protein-depleted cells, by measurements of total fluorescence of BODIPY-C12 in the whole area of cells. We found no substantial changes in the fluorescence of BODIPY-C12 among control, VPS13D-depleted, or ESCRT protein-depleted cells, suggesting that the increase in the size and number of LDs was not due to more FAs taken up by VPS13D, or ESCRT protein-suppressed cells.

The results were incorporated into the revised manuscript as Fig.6i, and were shown below.

4. Fig. 6l should include a column for cells depleted of VPS13D.

We thank reviewer for this comment. The result of VPS13D-depleted cells was incorporated into in Fig. 6k in the revised manuscript, and was shown below.

5. The data in Fig. 6b-d should include number of pmoles (or nmoles) of protein and lipids instead ug or ratios of fluorescence.

We thank reviewer for this comment. As suggested, the number of proteins and lipids was shown in Fig. 7c, d, e, and f in the revised manuscript, and was shown below.